https://doi.org/10.1038/s41467-019-10566-6　　**OPEN**

# Mechanistic basis of L-lactate transport in the SLC16 solute carrier family

Patrick D. Bosshart [1], David Kalbermatter [1,2], Sara Bonetti[1,2] & Dimitrios Fotiadis[1]

In human and other mammalian cells, transport of L-lactate across plasma membranes is mainly catalyzed by monocarboxylate transporters (MCTs) of the SLC16 solute carrier family. MCTs play an important role in cancer metabolism and are promising targets for tumor treatment. Here, we report the crystal structures of an SLC16 family homologue with two different bound ligands at 2.54 and 2.69 Å resolution. The structures show the transporter in the pharmacologically relevant outward-open conformation. Structural information together with a detailed structure-based analysis of the transport function provide important insights into the molecular working mechanisms of ligand binding and L-lactate transport.

---

[1] Institute of Biochemistry and Molecular Medicine, and Swiss National Centre of Competence in Research (NCCR) TransCure, University of Bern, CH-3012 Bern, Switzerland. [2] These authors contributed equally: David Kalbermatter, Sara Bonetti. Correspondence and requests for materials should be addressed to D.F. (email: dimitrios.fotiadis@ibmm.unibe.ch)

L-lactate is an important metabolite in health and disease[1]. Its stereoselective transport across plasma membranes is mainly catalyzed by monocarboxylate transporters (MCTs) of the SLC16 solute carrier family[2]. MCTs are predicted to adopt a protein fold that is characteristic for members of the major facilitator superfamily (MFS), which contain 12 transmembrane helices (TMs) arranged in two six-helix bundles. The bundles are connected by a long intracellular loop connecting TM6 and TM7, and related to each other by a pseudo-two-fold symmetry axis that is perpendicular to the membrane plane and along the substrate translocation pathway. As for other MFS transporters, the N- and C-termini are located intracellularly[2]. In TM8 an arginine residue is conserved in most SLC16 family members and proposed to be involved in binding of the carboxylate group of monocarboxylates[3,4]. Among the 14 mammalian SLC16 family members, MCT1-4 have been extensively studied and were identified as L-lactate:proton symporters[2]. An extracellular histidine residue is involved in pH regulation of MCT4 activity[5]. The unglycosylated MCT1-4 form heterodimers with the ancillary glycoproteins basigin (CD147) or embigin (GP70), which belong to the immunoglobulin superfamily and are responsible for proper trafficking to the plasma membrane[6,7]. However, they do not seem to be a prerequisite for transport function[8]. MCT1-4 exhibit a broad substrate specificity and are also involved in the transport of other monocarboxylic metabolites, e.g., pyruvate, ketone bodies and specific drugs[2]. Other members of the SLC16 solute carrier family encode high-affinity thyroid hormone (MCT8, SLC16A2) or aromatic amino acid (MCT10, SLC16A10) transporters, which mediate facilitated transport in contrast to proton-coupled transport[9]. While MCT1 (SLC16A1) is ubiquitously expressed, MCT2 (SLC16A7) is mainly found in the liver, kidney, testis, and in the brain[2]. MCT2 has a higher affinity for L-lactate ($K_m$ ~0.7 mM) than MCT1 ($K_m$ 3–5 mM) and both are responsible for the uptake of L-lactate into cells[2]. MCT3 (SLC16A8; $K_m$ ~ 6 mM[10]) is only expressed in the retinal pigment epithelium and the choroid plexus epithelia, where it serves as an L-lactate exporter. MCT4 (SLC16A3), which is strongly overexpressed in highly glycolytic and anaerobic tissues, has a low affinity for L-lactate ($K_m$ 20–35 mM) and is responsible for L-lactate export[2].

MCT1 (SLC16A1) and MCT4 (SLC16A3) are important in cancer metabolism. Certain tumor cells exhibit high glycolytic activity to cover the demand of ATP even under aerobic conditions ("Warburg effect"[11]). The glycolytic phenotype is maintained by transporting produced L-lactate out of the cell by MCT4. In cancer tissue, the acidification of the extracellular microenvironment resulting from co-transported protons is beneficial for angiogenesis, tumor cell proliferation, invasion and metastasis[11]. L-lactate that is exported by MCT4 serves as fuel for proliferating cancer cells, which import L-lactate by overexpressed MCT1[12]. Consequently, inhibition of MCT1 and MCT4 is a promising strategy for treating certain cancer types. In contrast to MCT1, no selective high-affinity inhibitor of MCT4 is available[13,14].

For the understanding of the molecular working mechanisms of L-lactate transporters such as MCT1 and MCT4, detailed structural information is indispensable. Furthermore, high-resolution structures are an important prerequisite for homology modeling and structure-based design of specific L-lactate transport inhibitors. Due to the lack of experimentally derived structures, the design of potent and highly selective inhibitors has been limited as well as the possibility to perform reliable ligand docking in silico.

Here, we present the structures of an L-lactate transporter. Two crystal structures were determined in the presence of the monocarboxylates thiosalicylate and L-lactate at 2.54 and 2.69 Å resolution. This bacterial L-lactate transporter (SfMCT) shares significant amino acid sequence identity with human MCT1 and MCT4. Importantly, the structures show SfMCT in an outward-open conformation. This conformational state with bound monocarboxylates is of high pharmacological relevance for the design of L-lactate transport inhibitors using SfMCT-based homology models of human MCT1 and MCT4. Furthermore, we performed structure-function studies to characterize substrate specificity and transport, and the substrate-binding site of SfMCT. The presented crystal structures and functional data provide important insights into the working mechanism of SLC16 solute carrier transporters at the molecular level.

## Results

**Functional characterization of SfMCT.** We have identified a transporter from *Syntrophobacter fumaroxidans* (SfMCT) that shares 25 and 27% amino acid sequence identity, and 51 and 57% sequence similarity with the human SLC16 family L-lactate transporters MCT1 and MCT4. The transport function of SfMCT was assessed by uptake of [$^{14}$C]L-lactate into SfMCT expressing *E. coli* JA202[15] (Supplementary Fig. 1a). SfMCT transports L-lactate with an average $K_m$ of $233 \pm 22$ µM (Fig. 1a). To determine the substrate specificity of SfMCT, we measured the ability of physiologically relevant mono-, di- and tricarboxylates at 1 mM concentration to compete with [$^{14}$C]L-lactate uptake (Fig. 1b, Supplementary Fig. 2). Among the tested linear unsubstituted monocarboxylates, only propionate and butyrate significantly reduced [$^{14}$C]L-lactate uptake (Fig. 1b) without being transported (Supplementary Fig. 1b). This indicates that besides a carboxylate group, an aliphatic chain consisting of at least three carbon atoms is required for significant competition. L-lactate, a chiral, α-hydroxyl substituted propionate, revealed the strongest competition among all tested physiologically relevant compounds and its D-isomer. Like MCT4[16], SfMCT has a clearly higher specificity for the α-hydroxyl monocarboxylate L-lactate than for the α-keto monocarboxylate pyruvate, which decreased the transport by only ~33% (Fig. 1b). SfMCT does not transport pyruvate as shown by the presented uptake experiment using radiolabeled pyruvate (Supplementary Fig. 1b). Positively charged amino groups are not tolerated at the α-carbon atom as indicated by the absence of any competition of [$^{14}$C]L-lactate uptake by L- and D-alanine. The monocarboxylic ketone bodies ((R)-3-hydroxybutyrate (3-HBA) and acetoacetate (ACAC)), the dicarboxylates oxalate and malonate as well as the tricarboxylate citrate reduced the [$^{14}$C]L-lactate uptake by only 5–14% (Fig. 1b). Comparison of the [$^{14}$C]L-lactate uptake competition by butyrate and the butyrate-derived monocarboxylic ketone bodies (i.e., 3-HBA and ACAC) demonstrates that hydroxyl or oxo groups are not tolerated at the β-carbon atom. SfMCT has a higher specificity for the dicarboxylate fumarate and its oxidized form succinate than for the shorter dicarboxylates oxalate and malonate. The uptake of [$^{14}$C]L-lactate was completely abolished by 20 µM of the protonophore carbonyl cyanide 3-chlorophenylhydrazone (CCCP) (Fig. 1b), while the addition of 20 µM valinomycin had no effect on the transport (Supplementary Fig. 1c). The transport of [$^{14}$C]L-lactate decreased with increasing pH values of the extracellular medium (Supplementary Fig. 1d). To determine if SfMCT is a proton-coupled L-lactate transporter, we studied the transport process using a micro pH electrode-based transport assay[17,18], where the electrode measures changes of the extracellular pH of a bacterial suspension. Addition of L-lactate to a suspension containing SfMCT-expressing *E. coli* JA202 resulted in an immediate pH increase (Supplementary Fig. 1e), although the injected L-lactate solution had a lower pH value than the bacterial suspension (L-lactate pH 6.5 vs bacterial suspension pH 6.7). Therefore, the observed pH increase reflects L-lactate

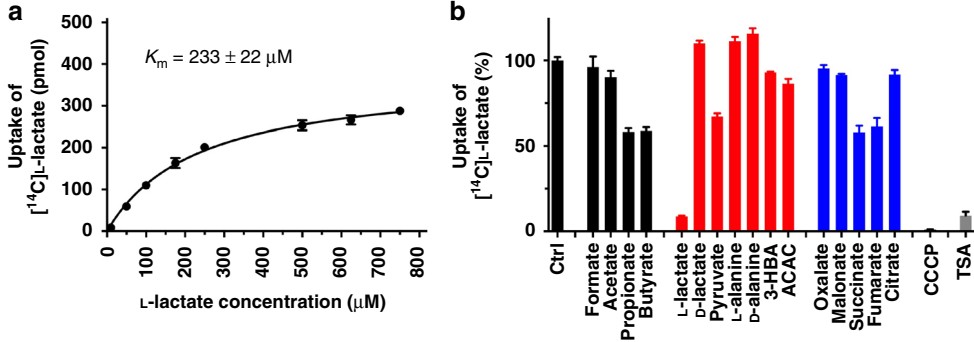

**Fig. 1** Functional characterization of SfMCT. **a** Concentration dependence of SfMCT-mediated [$^{14}$C]L-lactate uptake into *E. coli* JA202[15]. A representative saturation curve with Michaelis–Menten equation fit is shown. Data are represented as mean ± SEM from triplicates. An average $K_m$ of 233 ± 22 μM was calculated from three independent saturation experiments, each in triplicate. **b** Determination of the substrate specificity of SfMCT by [$^{14}$C]L-lactate transport competition assay (1 mM final concentration). Molecular structures of the used physiologically relevant compounds are shown in Supplementary Fig. 2. Residual uptake in the presence of competitor is normalized with respect to control samples without competitor (ctrl). Ketone bodies (R)-3-hydroxybutyrate and acetoacetate are abbreviated by 3-HBA and ACAC, respectively. Proton-dependence of the uptake was shown by 20 μM of carbonyl cyanide 3-chlorophenylhydrazone (CCCP). Thiosalicylate is abbreviated by TSA. Data are represented as mean ± SEM from three independent experiments, each in triplicate. If not visible, error bars are smaller than symbols. Source data are provided as a Source Data file

transport induced removal of protons from the extracellular solution. Addition of L-lactate to a suspension of vector-transformed bacteria induced a significantly slower and lower pH increase. These results demonstrate that L-lactate transport is proton-coupled and thus depends on a transmembrane proton gradient as observed for other L-lactate-transporting SLC16 family members[2]. Based on the obtained functional data (Fig. 1 and Supplementary Fig. 1a–e), SfMCT is a proton-coupled, L-lactate specific transporter.

**Overall structure of SfMCT.** The structure of SfMCT was solved at 2.54 Å resolution by single-wavelength dispersion (SAD) phasing from thiomersal-derivatized crystals (Supplementary Table 1). The quality of the obtained density map can be assessed in Fig. 2a. The correct register of the built model was confirmed by the anomalous difference densities of selenium atoms obtained of selenomethionine substituted SfMCT crystals (Supplementary Table 1 and Supplementary Fig. 3a). Crystals contain two SfMCT molecules in the asymmetric unit (Supplementary Fig. 4) that can be superimposed with a root-mean-square deviation of 0.261 Å (Supplementary Fig. 3b). SfMCT consists of 12 TMs with TM1-6 and TM7-12 arranged in an N- and C-terminal six-helix bundle. The bundles are related to each other by a pseudo-twofold symmetry axis, which is characteristic for the canonical MFS fold[19,20], and they can be superimposed with a root-mean-square deviation of 3.45 Å over 179 $C_\alpha$ atoms. The structure shows SfMCT in an outward-open conformation with a central, conical cavity open to the periplasmic side while the cytoplasmic side of SfMCT is closed (Fig. 2b, left panel). The SfMCT overall structure is comparable to those of MFS transporters that were solved in the same conformation[21]. The cavity, which is ~20 Å in depth, is surrounded by TM1, TM2, TM4 and TM5 of the N-terminal bundle, and TM7, TM8, TM10 and TM11 of the C-terminal bundle.

**Ligand-binding pocket description.** One molecule of the monocarboxylic ligand thiosalicylate (TSA), which is produced in situ during ethylmercury TSA derivatization of SfMCT crystals for phase determination, was found bound to SfMCT (Figs 2b and 3a–c; see also Supplementary Fig. 5a,b for electron density and omit maps). With an average inhibition constant ($K_i$) of 120 μM (Fig. 3d), SfMCT has a higher affinity for TSA than for its transported substrate L-lactate ($K_m$ = 233 μM) (Fig. 1a). In

contrast to L-lactate, TSA is not transported by SfMCT as determined by a micro pH electrode-based transport assay. Addition of TSA adjusted at pH 6.5 to suspensions of SfMCT-expressing or vector-transformed *E. coli* JA202 adjusted at pH 6.7 resulted in a decrease of the pH of the extracellular medium that recovered over time (Supplementary Fig. 1f). Thus, TSA did not lead to a net pH increase as observed for the transported L-lactate (Supplementary Fig. 1e) and previously for proton-coupled sugar transport through LacY[17]. Specific SfMCT residues interact through ionic, hydrogen bond and hydrophobic interactions with TSA constituting the SfMCT ligand-binding site (Fig. 3b, c). The carboxylate group of TSA interacts with the hydroxyl group of Y119 (TM4) and the Nη nitrogen atoms of the guanidinium group of R280 (TM8) (Fig. 3b, c). This positively charged residue is conserved in TM8 of most SLC16 family members[3,4]. L145 (TM5) and F335 (TM10) form an ~8 Å wide confinement that accommodates and orients the carboxylate group of TSA for proper binding to Y119 (TM4) and R280 (TM8). The thiol group of TSA is in hydrogen bonding distance to the hydroxyl group of Y331 (TM10) and 3.5 Å from the isobutyl group of L28 (TM1). These two residues trap TSA by partly obstructing the ligand-binding site (Fig. 4a, left panel). The van der Waals pore diameter along the substrate translocation pathway increases towards the periplasmic side of the protein as expected for an outward-open conformation (Fig. 4b). The side chains of F359 (TM11) and C362 (TM11) are within a distance of <4 Å from the benzene ring of TSA interacting via hydrophobic interactions. The binding of TSA demonstrates that residues from the N- and C-terminal parts of SfMCT are involved in ligand-binding interactions. To understand the binding mechanism of L-lactate that bound to the purified, detergent-solubilized transporter with a $K_d$ of 2.84 ± 0.15 mM (Supplementary Fig. 6a), we co-crystallized SfMCT with its transported substrate. Two molecules of L-lactate (i.e., Lac1 and Lac2) were found in the binding pocket (Fig. 5a–c; see also Supplementary Fig. 5c,d for electron density and omit maps). The carboxylate group of Lac1 interacts with the Nη nitrogen atoms of the guanidinium group of R280 (TM8), while the hydroxyl group is hydrogen bonded to the hydroxyl group of Y119 (TM4) (Fig. 5b, c). Furthermore, the carboxylate group of Lac1 is in hydrogen bonding distance to the hydroxyl group of Lac2. As observed for TSA, the Lac1 carboxylate group is sandwiched in the confinement formed by the side chains of L145 (TM5) and F335 (TM10). The phenyl-ring of F359 (TM11) points towards

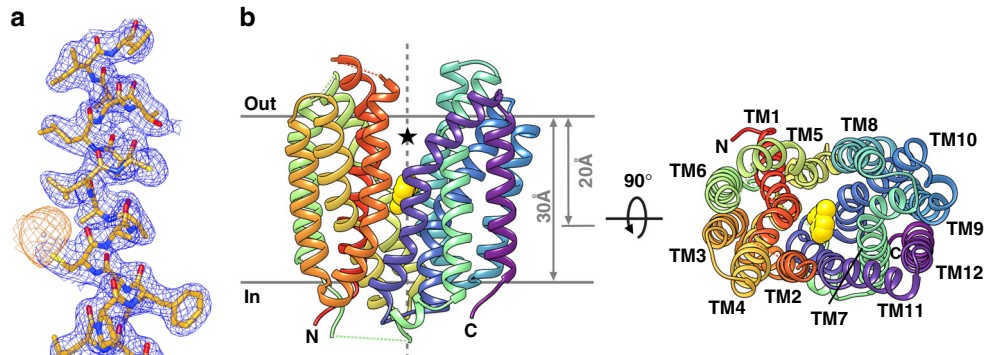

**Fig. 2** Electron density and overall structure of SfMCT. **a** Electron density map from the SfMCT crystal structure. A view of the final $2F_o$-$F_c$ electron density map of TM12 of SfMCT (colored in blue and contoured at 1.5 $\sigma$) is shown. The anomalous difference electron density (contoured at 3 $\sigma$) of one Hg atom is shown in orange. The SfMCT structure is displayed as stick model. **b** Overall structure of SfMCT in the outward-open conformation viewed in the plane of the membrane (left) and from the periplasm (right). One molecule of bound thiosalicylate is shown in yellow as space-filling model. The asterisk highlights the central, conical cavity that is open to the periplasmic side and the dark grey, vertical broken line the pseudo-twofold symmetry axis of SfMCT. The N- and C-termini are labeled. Parts of the termini and of the loops connecting TM1 and TM2, TM5 and TM6 as well as TM6 and TM7 could not be traced and are indicated by broken lines. Models are colored based on rainbow coloring scheme

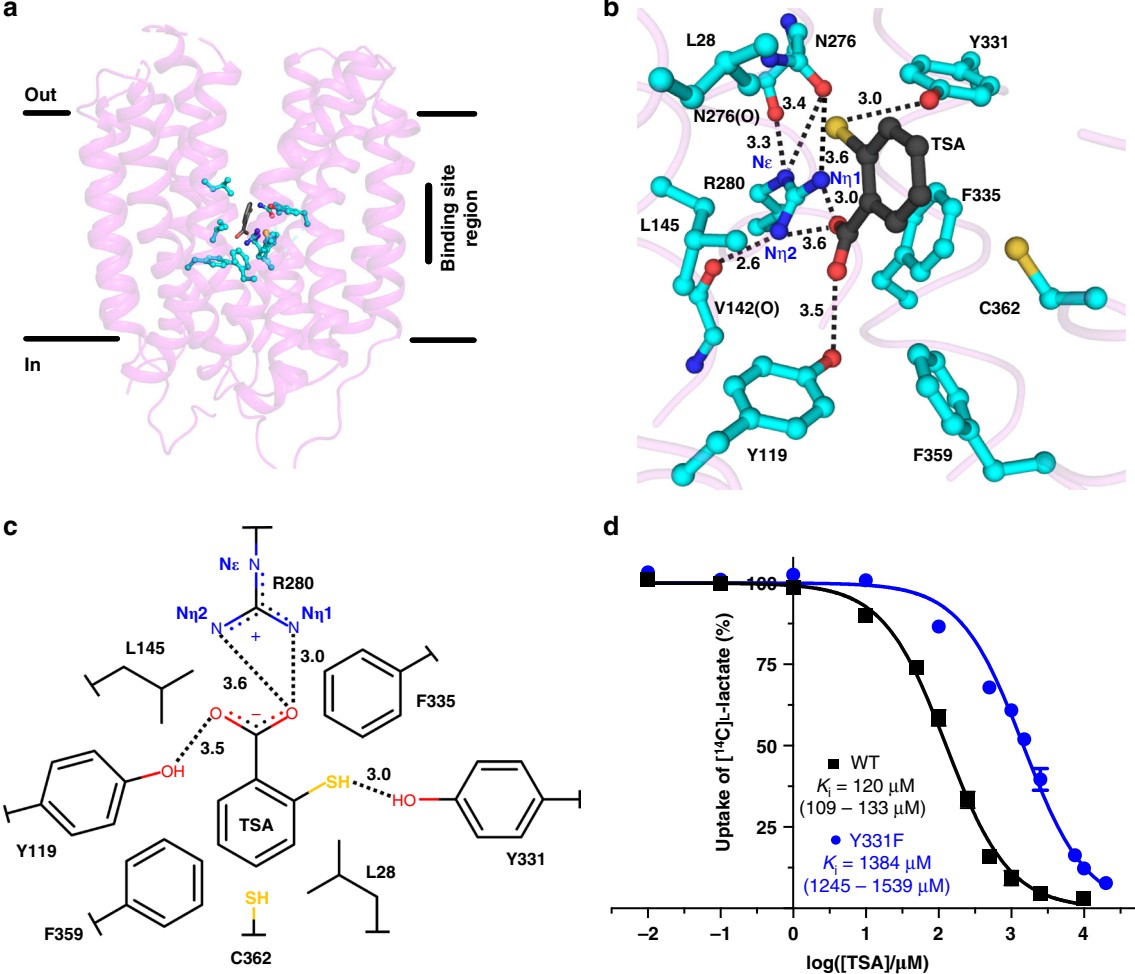

**Fig. 3** Binding pocket of SfMCT with bound TSA. **a** Overall structure of SfMCT in the outward-open conformation viewed in the plane of the membrane with indicated TSA (black) and binding site residues (cyan). **b** Binding mechanism of TSA to SfMCT. Residues within a distance of 4 Å from TSA, N276 and the peptide backbone of V142 are displayed as ball-and-stick model and are highlighted in cyan. **c** Two-dimensional schematic representation of key residues that are involved in TSA-binding. Distances in (**b**, **c**) are given in Ångström (Å). **d** $K_i$ determination of wild-type (WT) and Y331F SfMCT for TSA. The determined $K_i$ values are 120 μM (95% confidence interval: 109–133 μM) for WT and 1384 μM (95% confidence interval: 1245–1539 μM) for Y331F. Data are represented as mean ± SEM from three independent experiments, each in triplicate. Source data are provided as a Source Data file

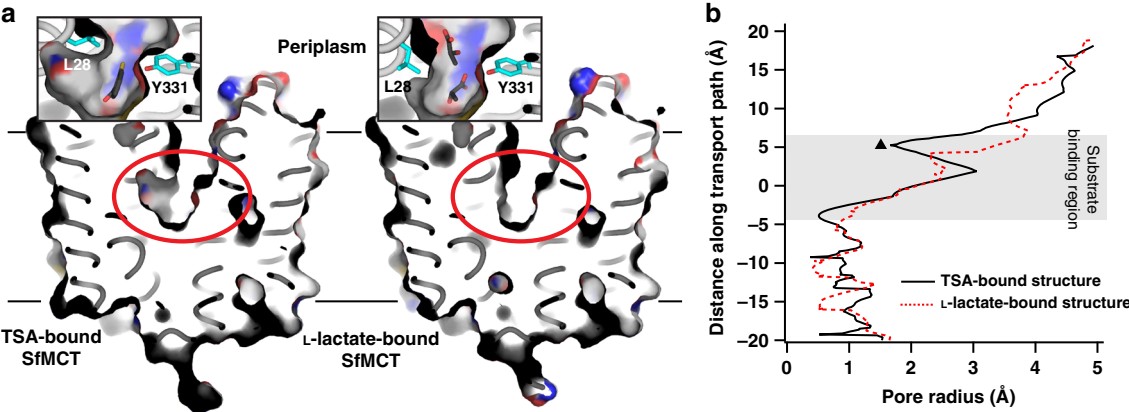

**Fig. 4** Analysis of van der Waals surfaces of SfMCT structures. **a** Cut-open views of structures of TSA (left) and L-lactate (right) bound SfMCT in the outward-open conformation. Insets show the obstruction site (red ellipse) between L28 (TM1) and Y331 (TM10) as well as the bound substrates TSA and L-lactate (insets). In the L-lactate bound SfMCT structure L28 (TM1) is rotated away from the obstruction site. **b** The van der Waals pore radius as a function of the distance along the transport path was computed using HOLE[50]. Plots represent the pore radii values of TSA (black) and L-lactate (red) bound SfMCT. The triangle highlights the obstruction site between L28 (TM1) and Y331 (TM10)

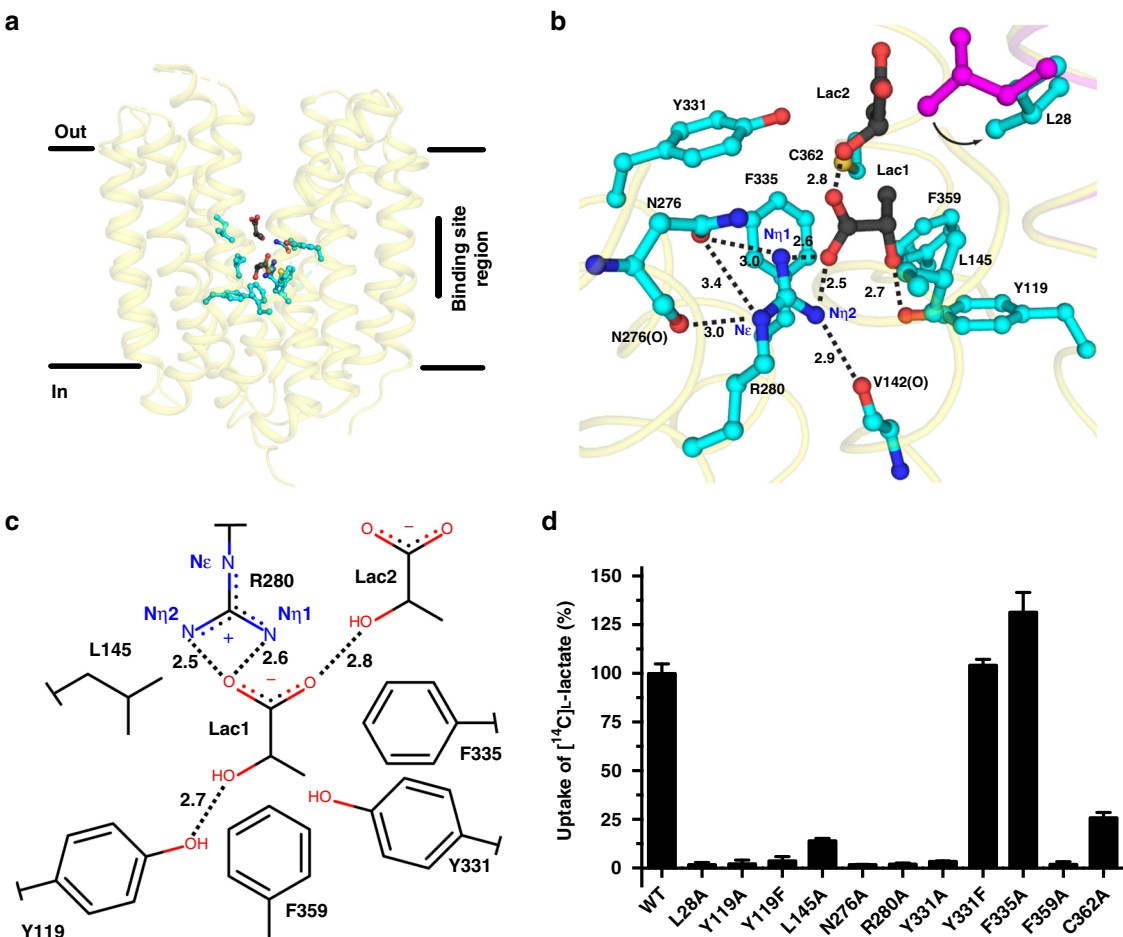

**Fig. 5** Binding pocket of SfMCT with bound L-lactate. **a** Overall structure of SfMCT in the outward-open conformation viewed in the plane of the membrane with indicated L-lactate (black) and binding site residues (cyan). **b** Binding mechanism of L-lactate to SfMCT. Residues within a distance of 4 Å from the two L-lactate molecules, N276, C362 and the peptide backbone of V142 are highlighted in cyan. L28 and the peptide backbone of TM1 from the SfMCT structure with bound TSA are shown in magenta. The conformational change of L28 occurring upon L-lactate binding is indicated by an arrow. **c** Two-dimensional schematic representation of residues that are involved in L-lactate-binding. Distances in (**b**, **c**) are given in Ångström (Å). **d** Determination of the role of TSA- and L-lactate binding residues on [14C]L-lactate transport using an in vivo uptake assay. Transport activities of SfMCT mutants were normalized with respect to wild-type protein (WT). Data are represented as mean ± SEM from at least three independent experiments, each in triplicate. Source data are provided as a Source Data file

Lac1 with a distance of 3.4 Å between the C4 carbon atom of the benzene ring and the methyl-group of Lac1. Most residues involved in ligand binding adopt similar conformations in the SfMCT structures with bound TSA and L-lactate. The exception is L28 (TM1), which is rotated away from the central cavity in the presence of L-lactate (Fig. 4a, right panel and Fig. 5b). This removes the binding site obstruction observed in the presence of TSA and generates space to accommodate Lac2 (Fig. 4). To investigate the mechanisms of TSA binding and L-lactate transport, we measured [14C]L-lactate uptake into bacteria expressing SfMCT versions harboring mutations of residues in the identified ligand-binding sites (Figs 3b and 5b). All variants were expressed as judged from Western blot analysis (Supplementary Fig. 7a). The replacement of R280 (TM8), which is involved in ligand binding (Figs 3b and 5b), by alanine (R280A) completely abolished L-lactate binding (Supplementary Fig. 6b) and [14C]L-lactate transport (Fig. 5d) as observed for MCT1[4]. N276 is conserved in TM8 of bacterial SLC16 homologues (Supplementary Fig. 8a). It interacts with R280 (TM8) via hydrogen bonds between its side chain amide group and the backbone carbonyl oxygen atom (N276(O) (TM8)), and the Nε and Nη1 nitrogen atoms of the guanidinium group of R280 (TM8) (Figs 3b and 5b, and Supplementary Fig. 8b). Furthermore, the Nη2 nitrogen atom of the guanidinium group of R280 (TM8) is hydrogen bonded to the backbone carbonyl oxygen of V142 (V142(O) (TM5)). This allows precise orientation of R280 (TM8) for proper interaction with the carboxyl group of ligands (Figs 3b and 5b). Replacing N276 (TM8) by alanine (N276A) led to an inactive transporter (Fig. 5d). Y119 (TM4) is involved in the binding of the carboxylate group of TSA (Fig. 3) and the hydroxyl group of Lac1 (Fig. 5). Neither Y119A nor the conservative mutation Y119F showed any [14C]L-lactate transport, which is in line with our observation that the hydroxyl group of L-lactate interacts with the hydroxyl group of Y119 (TM4) (Fig. 5b, c). L145 (TM5) and F335 (TM10) are close to R280 (TM8), and we have shown that they confine the substrate-binding site. Replacing these residues by alanine affected [14C]L-lactate transport: L145A (TM5) significantly reduced the uptake by 86%, while replacing the phenyl-ring of F335 (TM10) by alanine increased transport activity indicating that F335 might sterically interfere with substrate binding to assure substrate specificity. The alanine mutant Y331A is not functional but the conservative mutant Y331F transports [14C]L-lactate as wild-type (WT). While the hydroxyl group of Y331 (TM10) is not crucial for [14C]L-lactate transport, it is essential for TSA binding (Fig. 3). Y331F has a much lower affinity for TSA ($K_i = 1384\,\mu M$) compared to WT SfMCT ($K_i = 120\,\mu M$) (Fig. 3d). Substitution of F359 and C362 (TM11) by alanine led to complete transport abrogation (F359A) and a low residual transport activity (26%, C362A), respectively (Fig. 5d).

**Proton binding site of SfMCT and pH dependence of L-lactate transport.** L-lactate transport through SfMCT depends on a transmembrane proton gradient (Fig. 1b and Supplementary Fig. 1d). Since protons are co-transported with L-lactate (Supplementary Fig. 1e), a binding site for the protons must be present in SfMCT. In MCT4 a histidine residue that is accessible from the extracellular side is involved in the pH-dependent regulation of the transport activity[5]. In SfMCT there are only two histidine residues. H250 is intriguing, because it is located at the end of a transmembrane domain (TM7) near the periplasm (Fig. 6a,b) where it can bind protons from the extracellular space. The second histidine residue is found intracellularly at the second to last position of the C-terminus (H411), thus not being exposed to the periplasm. Due to the pKa value of histidine, H250 (TM7) could act as a titratable proton acceptor under physiological conditions

as observed for MCT4[5]. The side chain of H250 (TM7) is exposed to the central cavity and oriented by a hydrogen bond between its imidazole ring and the hydroxyl group of Y383 (loop between TM11 and TM12), which is conserved in SfMCT and MCT1-4 (Fig. 6b and Supplementary Fig. 9). Positively (R256 (TM7) and K377 (TM11)) and negatively (D257 (TM7) and D378 (TM11)) charged residues are arranged in an alternating fashion above H250 (TM7) and Y383 (loop between TM11 and TM12), and point towards the central cavity that is open to the periplasm. To investigate the role of H250 (TM7) and the other residues of the potential proton-binding site (Fig. 6a, b), we measured the transport function of corresponding mutants. All variants were expressed as judged from Western blot analysis (Supplementary Fig. 7b). While [14C]L-lactate transport through H250A was significantly reduced to only ~12%, D257A, D378A and Y383A did not transport any substrate (Fig. 6c). Removal of the positively charged amino acids R256A and K377A led to an increased negative electrostatic surface charge in the potential proton binding site (Supplementary Fig. 10). [14C]L-lactate transport through these mutants was comparable with transport through WT SfMCT (Fig. 6c). The negative charge density was further increased by replacing R256 (TM7) and K377 (TM11) by aspartate (i.e., R256D and K377D). [14C]L-lactate transport through R256D was ~2.5 fold increased, while uptake through K377D was similar to WT SfMCT. The observed increased transport through R256D cannot be attributed to a higher expression level (Supplementary Fig. 7b). Based on the observed efficient L-lactate transport of the R256A/D and K377A/D mutants, we conclude that the positive charges of R256 (TM7) and K377 (TM11) are not crucial for initial binding of the negatively charged L-lactate substrate on the extracellular surface of SfMCT and its transfer towards the substrate-binding site. On the other hand, the fact that removal of negatively charged residues (D257A and D378A) resulted in inactive transporters suggests their involvement in proton binding. Next, we replaced the titratable H250 (TM7) by a phenylalanine to introduce a non-titratable side chain of comparable steric demand that cannot be involved in hydrogen bond interactions. This H250F mutation resulted in an inactive transporter. A possible explanation for this observation is that phenylalanine cannot be oriented appropriately due to the missing hydrogen bond interaction with the hydroxyl group of Y383 (loop between TM11 and TM12). This interaction with the imidazole ring of H250 (TM7) is functionally important as shown by the strongly reduced transport activity of the conservative mutant Y383F (Fig. 6c).

We investigated the pH-dependence of [14C]L-lactate transport through WT SfMCT, the fully functional alanine mutants (R256A and K377A) and the charge-inversion mutant R256D, which revealed a significantly higher activity than WT SfMCT. All tested mutants and WT SfMCT showed higher transport activities at lower than at higher pH-values as expected for a proton-dependent transporter (Fig. 6d). This supports our previous observations of proton-coupled L-lactate transport (Supplementary Fig. 1e) and the completely abolished substrate transport by CCCP (Fig. 1b). Under basic conditions R256D had a similar uptake as WT SfMCT under acidic conditions (Fig. 6e). Transport activities of R256A, R256D and K377A were significantly higher at alkaline pH-values than for the virtually inactive WT SfMCT (Fig. 6d).

## Discussion

We have presented the structures of an L-lactate transporter (SfMCT) with two different bound ligands. Both structures show SfMCT in the pharmacologically relevant outward-open conformation, which is an essential prerequisite for the design of

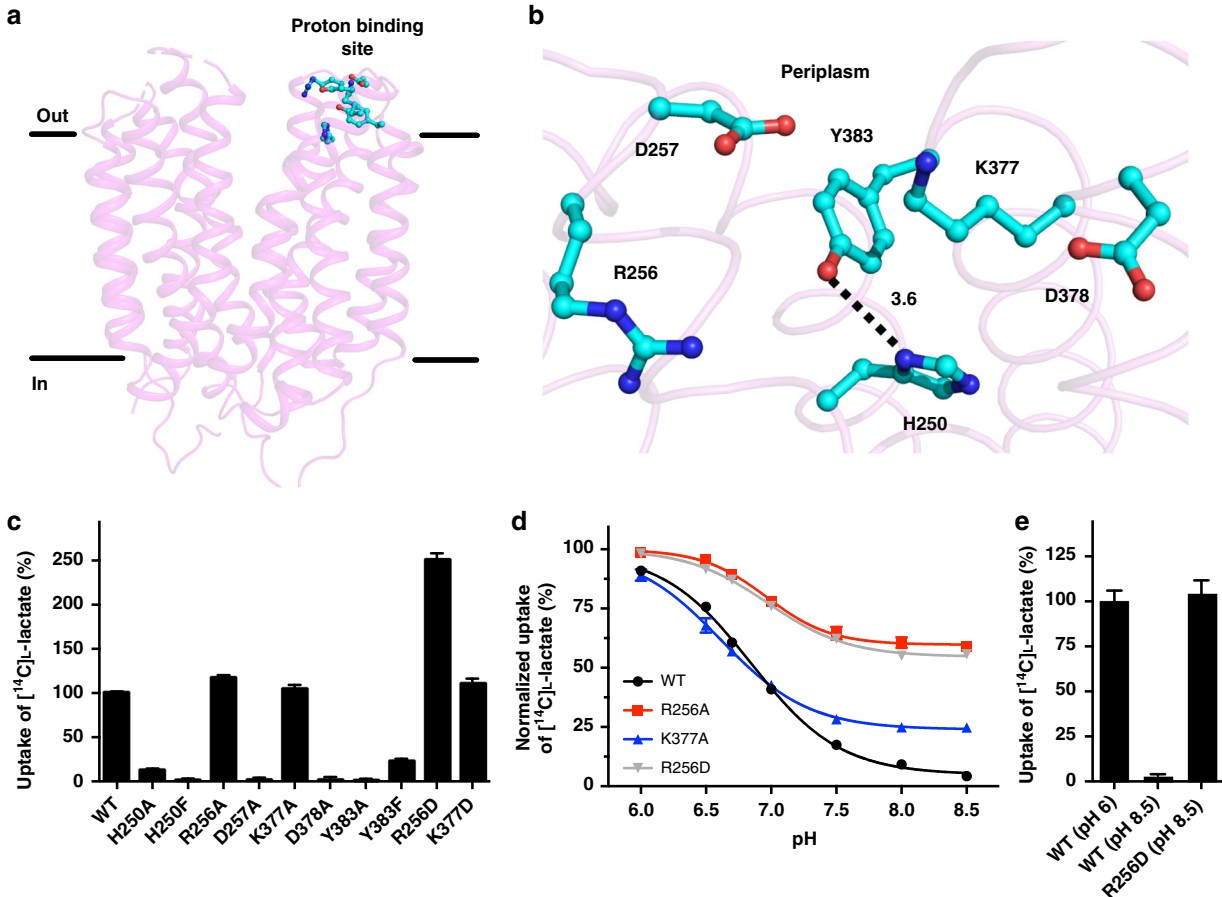

**Fig. 6** Proton binding site of SfMCT and pH dependence of L-lactate transport. **a** Overall structure of SfMCT in the outward-open conformation viewed in the plane of the membrane with indicated proton binding site residues (cyan). **b** Structure of the proton binding site region comprising H250 (TM7), R256 (TM7), D257 (TM7), K377 (TM11), D378 (TM11) and Y383 (loop between TM11 and TM12). **c** Determination of the role of potential proton binding site residues on [$^{14}$C]L-lactate transport using an in vivo uptake assay. Transport activities of SfMCT mutants were normalized with respect to wild-type (WT). Data are represented as mean ± SEM from three independent experiments, each in triplicate. **d** pH-dependence of proton binding site mutants. Each trace represents the average of three independent experiments that were individually normalized by fitting a sigmoidal curve to the data points and setting the fitted upper plateau value to 100% [$^{14}$C]L-lactate uptake (see Methods section for details on the normalization procedure). Data are represented as mean ± SEM from three independent experiments, each in triplicate. If not visible, error bars are smaller than symbols. **e** Comparison of [$^{14}$C]L-lactate transport through WT SfMCT at pH 6, and WT and R256D at pH 8.5. The uptake through WT at pH 6, and WT and R256 at pH 8.5 were normalized with respect to the [$^{14}$C]L-lactate uptake through WT at pH 6. Data are represented as mean ± SEM from three independent experiments, each in triplicate. Source data are provided as a Source Data file

L-lactate transport inhibitors using SfMCT structure-based homology models of human MCT1 and MCT4. Homology models of rat MCT1 and MCT4 have been previously published[7,22,23]. These models are based on the inward-open structure of the glycerol-3-phosphate:phosphate antiporter GlpT[19], which belongs to a different MFS subfamily (i.e., the organophosphate:phosphate antiporter family), and GlpT shares low sequence identity with rat MCT1 and MCT4 (i.e., 18.5 and 17.3%, respectively). Importantly, functionally relevant residues are not conserved between GlpT and MCTs. Secondary structure analysis of the outward-open rat MCT1 homology model[23] revealed that many transmembrane spanning domains were fragmented consisting of relatively short α-helices, and a considerable amount of hydrogen-bonded turns, bends and 3$_{10}$-helices[24–26] (Supplementary Fig. 11a). Such a high fraction of non-α-helical structural elements is not observed in transmembrane domains of prokaryotic and eukaryotic MFS transporters. Furthermore, TM2 is distorted and no α-helical structure was detected in the analysis (Supplementary Fig. 11a). Overlaying the rat MCT1 model with our experimental SfMCT structure resulted in a high root-mean-square deviation of 6.44 Å (over 1882 atoms) and revealed that the topologies are roughly comparable, but the secondary structure is significantly different. Furthermore, the structural alignment has uncovered significant differences between the locations and orientations of functionally relevant key residues, which are responsible for the binding of TSA and L-lactate (Supplementary Fig. 11b). In summary, incorrect and unprecise locations/orientations of functional key residues in the rat MCT1 homology model preclude the reliable use of structure-based applications, e.g., molecular dynamics simulations and structure-based drug design.

Our ligand-bound SfMCT structures and functional studies provide a framework for understanding the molecular working mechanisms of substrate binding of SLC16 transporters. SfMCT shares significant amino acid sequence identity and similarity with human MCT1 and MCT4. It is a valid model to study their transport function since SfMCT also transports L-lactate in a stereospecific and proton-dependent manner. Furthermore, the affinity of SfMCT for L-lactate is higher than for pyruvate and monocarboxylic ketone bodies as observed for MCT4[2]. Therefore,

the oxidation state and geometry of the α-carbon atom (e.g., *sp*3 for L-lactate vs *sp*2 hybridization for pyruvate) affects the substrate specificity of SfMCT similar to mammalian MCT1 and MCT4. Since SfMCT does not tolerate an amino group at the α-carbon atom of a substrate, this excludes amino acids and their derivatives as substrates. Therefore, SfMCT does not represent an optimal model for the well-studied SLC16 thyroid hormone (MCT8) and aromatic amino acid (MCT10) transporters[2]. These two SLC16 family members are also different from MCT1-4 because they are not associated to ancillary proteins[9]. The structures of TSA- and L-lactate-bound SfMCT demonstrate that the conserved arginine residue in TM8 is essential for monocarboxylate binding. This key residue is also crucial for function in certain mammalian SLC16 family members[4,27]. Not even a conservative mutation to lysine, which is a primary amine and shorter than arginine, was able to maintain MCT1 function[22]. Mutation of this arginine to glutamine resulted in an inactive version of human MCT1, which is associated with ketoacidosis[28]. The resulting non-functional MCT1 cannot bind and transport monocarboxylic ketone bodies into target cells, since the positively charged guanidinium group is essential for substrate binding. However, not only the presence of the guanidinium group of R280 is important for substrate binding. The correct positioning of R280 (TM8), which is maintained by hydrogen bonds to the side chain of the conserved N276 (TM8) and to peptide backbone atoms (Supplementary Fig. 8b), is also a prerequisite for the efficient binding of the substrate carboxylate group (Figs 3 and 5). Based on our structures and functional data, Y331 (TM10) is located at a strategic position and it seems to be involved in ligand-binding specificity. This is highlighted by the fact that removal of its hydroxyl group drastically reduced the affinity for TSA binding, while L-lactate was still properly transported by Y331F. This mutant reflects the situation in human MCT1 (Supplementary Fig. 9) that efficiently transports L-lactate into cells. However, replacing the aromatic side chain by alanine resulted in an inactive SfMCT transporter similar to mammalian MCTs[4,29]. Interestingly, a phenylalanine to cysteine exchange at the corresponding position in MCT1 resulted in a gain of function mutant. This mutant did not transport L-lactate, but catalyzed the transport of the slightly larger substrate mevalonate, which is not transported by WT MCT1[30]. The confinement formed by L145 (TM5) and F335 (TM10) accommodates the carboxylate groups of TSA and L-lactate. Together with Y119 (TM4), these residues seem to be involved in a mechanism to regulate the specificity and transport efficiency of L-lactate in SfMCT. Replacing sterically demanding residues (e.g., leucine and phenylalanine) generates more space for the carboxylate to reorient such that bulkier substrates are bound. This has been shown when the phenyl-ring of F335 (i.e., F335A) was removed resulting in a higher specificity for the bulkier ketone body ACAC (Supplementary Fig. 1g).

The presented functional characterization has shown that L-lactate transport through SfMCT is coupled to the co-transport of protons, which requires proton binding. We have identified a proton binding site consisting of residues at the periplasmic ends of TM7 (H250, R256 and D257) and TM11 (D378), and in the loop between TM11 and TM12 (Y383) (Fig. 6a, b). The two negatively charged aspartate residues (D257 (TM7) and D378 (TM11)) might act as antennas for protons from the periplasm, and thus they are essential for transport as shown experimentally (Fig. 6c). Their relevance in proton-dependent L-lactate transporters is underlined by their conservation in SfMCT and human MCT2-4 (Supplementary Fig. 9). The response of [¹⁴C]L-lactate transport on extracellular pH changes was affected by the positively charged amino acid R256 (TM7) and substitutions of this residue. Its charge-neutralization (R256A) or -inversion (R256D)

rendered the corresponding SfMCT mutants active even at basic pH values, where proton-dependent transporters are usually inactive. The higher activity of these two SfMCT mutants under alkaline conditions suggests that the coupling between L-lactate and proton transport is altered upon removal of the positive charge at this position. The increased negative charge density of the proton binding site associated with R256A or R256D raises the affinity for positively charged protons such that a lower extracellular proton concentration (i.e., basic pH-values) is sufficient for driving efficient transport. This is highlighted by the fact that R256D showed similar uptake under basic pH conditions as WT SfMCT under acidic pH conditions (Fig. 6e). However, simply increasing the negative charge density of the proton binding site is not sufficient to improve activity as shown by the unaffected transport through K377D (Supplementary Fig. 10g and Fig. 6c). Therefore, the correct position of the additional negative charge is crucial as demonstrated by the effect of the charge-inversion R256D. Under basic conditions the positively charged residues R256 (TM7) and K377 (TM11) might interact with the two negatively charged aspartate residues via unresolved water molecules. Consequently, protonation of antenna residues would lead to an increase of the local positive charge density. In the absence of negatively charged amino acids (i.e., D257A and D378A), the resulting net positive charge density of R256 (TM7) and K377 (TM11) might reduce the accessibility of H250 (TM7) for protonation due to electrostatic repulsion. In WT SfMCT, protons that were captured by the proton antennas (i.e., D257 (TM7) and D378 (TM11)) might be transferred to H250 (TM7), which can sense extracellular pH due to its appropriate location near the periplasm (Fig. 6a, b). In MCT4, an essential histidine residue involved in pH regulation of transport activity was identified in a similar region like the proton binding site in SfMCT[5]. We propose that protonation of the imidazole ring leads to breaking of the weak hydrogen bond between the conserved and for [¹⁴C]L-lactate transport essential Y383 (Fig. 6b and Supplementary Fig. 9) and H250. This would allow adopting other conformers of the side chains of H250 and Y383, thus allowing proton transfer towards the cytoplasmic part of SfMCT. However, transfer of captured protons from the initial proton antennas to H250 might also occur via water molecules in the form of hydronium intermediates, which could not be resolved in the presented structures at given resolutions.

In summary, we reported two structures of the bacterial SLC16 homologue SfMCT in complex with the monocarboxylates TSA and L-lactate. The presented structural and functional data have provided important insights into the molecular working mechanism of L-lactate transporters.

## Methods

**Cloning**. The gene of a major facilitator superfamily-type transporter from *Syntrophobacter fumaroxidans* (SfMCT, UniProt ID code A0LNN5) was synthesized codon-optimized for expression in *E. coli* (GenScript, Supplementary Table 2). SfMCT was ligated into the pZUDF21-rbs-3C-10His vector[31] using 5'-HindIII and 3'-XhoI restriction sites for overexpression, or into the pEXT20 vector[32] using 5'-EcoRI and 3'-XhoI restriction sites for functional studies. The resulting constructs (pZUDF21-rbs-SfMCT-3C-10His and pEXT20-SfMCT-3C-10His) contained a C-terminal human rhinovirus 3C (HRV3C) protease cleavage site and a decahistidine-tag (His-tag). Site-directed mutagenesis was done using a proofreading polymerase (KAPA HiFi DNA Polymerase, KAPABiosystems) and the primers listed in Supplementary Table 3. DNA sequences were verified by gene sequencing (Microsynth).

**Expression and membrane preparation**. SfMCT was expressed in *E. coli* BL21 (DE3)pLysS grown at 37 °C in Luria Bertani (LB) medium supplemented with antibiotics (100 µg/ml ampicillin and 36 µg/ml chloramphenicol) at 180 r.p.m. in an incubator shaker (Multitron, Infors HT). Protein expression was induced at OD₆₀₀ ~0.9 by the addition of isopropyl-β-D-thiogalactopyranoside (IPTG) to a final concentration of 0.25 mM. After four hours, cells were harvested by centrifugation (10,000 × *g*, 6 min, 4 °C), resuspended in lysis buffer (45 mM Tris-HCl

(pH 8), 450 mM NaCl, 4 °C) and pelleted again (10,000 × g, 25 min, 4 °C). The final cell pellet was resuspended in lysis buffer. Bacteria were disrupted using an M-110P Microfluidizer (Microfluidics) operated at 1,500 bar during six passages. Unlyzed bacteria were removed by low-speed centrifugation (10,000 × g, 10 min, 4 °C). The supernatant was subjected to ultracentrifugation (200,000 × g, 90 min, 4 °C) to isolate bacterial membranes. The resulting pellet was resuspended in lysis buffer, homogenized and subjected to ultracentrifugation (200,000 × g, 90 min, 4 °C). Finally, membranes were resuspended and homogenized in buffer (20 mM Tris-HCl (pH 8), 150 mM NaCl, 10% (v/v) glycerol) at 100 mg/ml and stored at −80 °C.

**Expression of L-selenomethionine-labeled SfMCT.** For the expression of L-selenomethionine-labeled SfMCT (SfMCT[SeMet]), an overnight culture of E. coli BL21(DE3)pLysS transformed with pZUDF21-rbs-SfMCT-3C-10His was grown at 37 °C in LB medium supplemented with antibiotics (100 μg/ml ampicillin and 36 μg/ml chloramphenicol) at 180 r.p.m in an incubator shaker (Multitron, Infors HT). The culture was pelleted (10,000 × g, 6 min, 4 °C) and resuspended in minimal medium (33.7 mM Na₂HPO₄, 22 mM KH₂PO₄, 9 mM NaCl, 10 mM (NH₄)₂SO₄, 1 mM MgSO₄, 0.1 mM CaCl₂, 20 mM D-glucose, 1 mg/l thiamine-hydrochloride and 1 μM FeSO₄). Washed bacteria were diluted 1:100 in minimal medium supplemented with 100 μg/ml ampicillin and grown to OD₆₀₀ ~0.6 at 37 °C and 180 r.p.m. in an incubator shaker (Multitron, Infors HT). An L-amino acid mixture (lysine (100 mg/l), phenylalanine (100 mg/l), threonine (100 mg/l), isoleucine (50 mg/l), leucine (50 mg/l), valine (50 mg/l)) and L-selenomethionine (60 mg/l) were added to the culture to inhibit the L-methionine synthesis pathway. The culture was incubated for 30 min at 37 °C before inducing SfMCT[SeMet] expression by addition of IPTG to a final concentration of 0.25 mM. Further processing of the expression culture, membrane isolation and purification followed the same procedure as described for unlabeled SfMCT.

**Purification.** All purification steps were performed at 4 °C. Membranes were solubilized by gentle stirring in solubilization buffer (20 mM Tris-HCl (pH 8), 150 mM NaCl, 10% (v/v) glycerol, 4% (w/v) n-nonyl β-D-glucopyranoside (NG, Glycon Biochemicals GmbH)) for 2 h. Subsequently, solubilized membranes were clarified by ultracentrifugation (200,000 × g, 30 min, 4 °C). The supernatant was diluted with an equal volume of detergent-free solubilization buffer supplemented with 5 mM L-histidine. Solubilized membranes were then incubated with nickel-nitrilotriacetate resin (Ni-NTA; ProteinIso) (1 ml resin bed volume for solubilized membranes from one liter expression culture) under gentle stirring. After 2 h, the resin was transferred into a column using a peristaltic pump operated at 4 ml/min. The resin was washed with 25 column volumes of washing buffer (20 mM Tris-HCl (pH 8), 150 mM NaCl, 5% (v/v) glycerol, 5 mM L-histidine, 0.4% (w/v) NG). SfMCT was eluted from the column by incubating the resin with His-tagged HRV3C on a rotational shaker for ~16 h. Co-eluted HRV3C and undigested SfMCT were removed by an additional Ni-NTA purification.

**Crystallization.** Purified SfMCT or SfMCT[SeMet] was concentrated to 8 mg/ml using a 50,000 Da molecular weight cutoff concentration device (SARTORIUS Stedim Biotech, Vivaspin 2). Aggregated protein was removed by ultracentrifugation (150,000 × g, 30 min, 4 °C). The protein was crystallized in the sitting-drop vapor-diffusion method by mixing concentrated protein with reservoir solution (50 mM HEPES-NaOH (pH 7), 5 mM ZnBr₂, 30% (v/v) Jeffamine ED-2003) using a Mosquito Crystal Robot (TTP Labtech). Crystals appeared after one day of incubation at 18 °C and reached maximal size after one week. Before X-ray analysis, crystals were flash frozen in liquid nitrogen. For mercury derivatization, crystals were incubated in reservoir solution supplemented with 0.5 mM ethyl-mercury thiosalicylate and 0.5% (w/v) NG. For co-crystallization with L-lactate, all purification buffers were supplemented with 10 mM sodium L-lactate.

**Data collection and structure determination.** All data sets were collected on frozen crystals at the X06SA (PXI) beamline of the Swiss Light Source (Paul Scherrer Institute, Villigen, Switzerland) using an EIGER 16M detector (Dectris). Three data sets were collected at low-dose per frame and high-redundancy for phasing. For SeMet two data sets and for the L-lactate structure four data sets from different crystals were collected. The data sets were indexed and integrated with XDS[33], then merged using BLEND[34] of the CCP4 program suite[35] without truncation of the resolution. The high-resolution data set was collected from a single crystal at high-dose per frame (Supplementary Table 1). The data were indexed and integrated with XDS[33] without truncation of the resolution. Scaling and averaging of symmetry-related intensities for all data sets were performed by aP_scale[36] with truncation of the data at the best high-resolution along h, k or l axis determined by AIMLESS[37]. Due to the anisotropic nature of the diffraction data, the STARANISO software (http://staraniso.globalphasing.org/) was applied. This program performs an anisotropic cut-off of merged intensity data to perform Bayesian estimation of structure amplitudes and to apply an anisotropic correction to the data. For phasing and generation of an initial SfMCT model, the SAD method was applied using the CRANK2[38] automated structure solution pipeline running SHELX[39]/D[40], REFMAC5[41], Parrot[42] and Buccaneer[43] in the CCP4 program suite[35]. In CRANK2, phases were extended to the diffraction limit of the high-resolution data set. To obtain a complete structure, iterative refinement and model building were

performed by phenix.refine[44] and Coot[45] using the high-resolution data set, respectively. The structure of SfMCT with bound L-lactate was obtained by molecular replacement with Phaser[46] using the SfMCT structure with bound TSA as search model. The final structures were obtained after multiple rounds of model building with Coot[45] and refinement with phenix.refine[44]. For all the refinements, XYZ coordinates, individual B-factors, occupancies and TLS strategies were applied. The TLS groups were automatically assigned using Phenix. Full data collection, processing and refinement statistics can be found in Supplementary Table 1. Figures representing structural information were prepared using Chimera[47] or PyMol (The PyMol Molecular Graphics System; Schrödinger).

**Calculation of electrostatic surface potentials of SfMCT variants.** The electrostatic surface potentials of SfMCT and SfMCT variants were calculated using the APBS plugin of PyMol. SfMCT variants were generated in silico using the mutagenesis tool of PyMol. The resulting APBS-calculated electrostatic surface potentials were presented using PyMol.

**Uptake experiments.** Overnight precultures of E. coli JA202 (MC4100 glcA::cat lldP::kan)[15], a strain that lacks the endogenous L-lactate transporters LldP and GlcA, transformed with empty vector (pEXT20-3C-10His) or plasmids carrying SfMCT and SfMCT mutants (pEXT20-SfMCT-3C-10His) were diluted 1:200 into LB-medium supplemented with 100 μg/ml ampicillin. The cultures were grown at 37 °C and 180 r.p.m. in an incubator shaker (Multitron, Infors HT). Protein expression was induced at OD₆₀₀ ~0.5 by addition of IPTG to a final concentration of 0.25 mM. After 4 h, bacteria were pelleted (5,200 × g, 10 min, room temperature) and resuspended in uptake buffer (20 mM Bis-Tris propane-HCl (pH 6.7), 250 mM KCl) to a bacteria density of OD₆₀₀ 12. Transport experiments were done in a reaction volume of 50 μl that included 20 μl of cell-suspension (2.4 × 10⁸ bacteria), 10 μl of substrate master mix (67 μM sodium L-lactate spiked with [14 C(U)] L-lactic acid sodium salt ([¹⁴C]L-lactate, American Radiolabeled Chemicals) to a specific activity of 0.15 Ci/mmol) and 20 μl of competitor solution. Furthermore, [3-3 H] propionic acid sodium salt ([³H]propionate, American Radiolabeled Chemicals, specific activity 30 Ci/mmol), n-[2,3-3 H] butyric acid sodium salt ([³H] butyrate, American Radiolabeled Chemicals, specific activity 120 Ci/mmol), and [1–14 C] pyruvic acid sodium salt ([¹⁴C]pyruvate, specific activity 0.055 Ci/mmol American Radiolabeled Chemicals) were also used for transport experiments. For $K_m$ determination, [¹⁴C]L-lactate was diluted to a specific activity of 0.0008 Ci/mmol. Uptake experiments were performed in 2 ml reaction tubes (Eppendorf) at 30 °C under agitation (1,000 r.p.m., Thermomixer compact, Eppendorf). Reactions were stopped after 30 min by addition of 900 μl stop buffer (20 mM HEPES-NaOH (pH 7.5), 150 mM NaCl). Bacteria were immediately pelleted by centrifugation (21,000 × g, 4 min, room temperature) and washed two times with 900 μl stop buffer. Finally, bacteria were lyzed in 50 μl 5% (w/v) sodium dodecylsulfate and transferred into a white 96-well plate (Optiplate, PerkinElmer). 150 μl of scintillation cocktail (MicroScint 40, PerkinElmer) were added before measuring each reaction for 2 min with a scintillation counter (TopCount NXT, PerkinElmer). For data analysis, the signal of the empty vector was subtracted from the transporter signal. The pH-dependence of [¹⁴C]L-lactate transport through SfMCT variants was measured in Bis-Tris propane based uptake buffers (20 mM Bis-Tris propane-HCl, 250 mM KCl) that were adjusted to the desired pH values using HCl. pH-dependence curves were individually processed by fitting a sigmoidal curve to the raw data of each individual experiment, i.e., counts-per-minute (cpm) vs pH. Cpm values of each experiment were normalized with respect to the determined upper plateau value, i.e., the fitted upper plateau value corresponds to 100%. The arithmetic average of three independent experiments was calculated. Three independent and normalized pH-dependence curves were then averaged. For data analysis Prism6 (GraphPad Software) was used. The expression of SfMCT variants in E. coli JA202 (MC4100 glcA::cat lldP::kan)[15] was verified by Western blotting. Equal amounts of bacteria expressing SfMCT variants were loaded on 14% SDS/poly-acrylamide gels. His-tagged transporters were detected using an anti-His₅ (Qiagen, catalogue number 34660, primary antibody) at a dilution of 1:3,000 and a goat anti-mouse IgG (H + L) HRP conjugate antibody (Biorad, catalogue number 172-1011, secondary antibody) at a dilution of 1:2,500.

**Micro pH electrode-based transport measurements.** A measuring set up as previously reported[18] was used for micro pH electrode-based transport measurements. SfMCT was expressed in E. coli JA202 (MC4100 glcA::cat lldP::kan)[15] as described for uptake experiments. After expression, bacteria were pelleted (5,200 × g, 10 min, room temperature), washed twice in assay solution (250 mM KCl, 1 mM MgSO₄, 2 mM CaCl₂; 5,200 × g, 10 min, room temperature) and finally resuspended in assay solution to a bacterial density of OD₆₀₀ 15. Compounds that were assayed for proton-coupled transport were dissolved in assay solution and the pH was adjusted to 6.5. 800 μl of bacteria suspension were transferred into a 2 ml reaction tube (Eppendorf) and the pH value of the bacterial suspension (extracellular medium) was measured using a micro pH electrode with integrated temperature sensor (InLab Micro Pro, Mettler Toledo) under constant stirring. The pH of the bacterial suspension was adjusted to pH 6.7 using potassium hydroxide and hydrochloric acid. The SevenCompact pH-meter (Mettler Toledo), together with LabX direct pH 2.3 software (Mettler Toledo), were used for automatic recording in

15 s intervals. Measurements were initiated by adding 10 mM of the corresponding compound to the bacterial suspension. All experiments were performed at 24 °C.

**Microscale thermophoresis binding assay.** Binding of L-lactate to purified, detergent-solubilized SfMCT and SfMCT-R280A was measured with the micro-scale thermophoresis (MST) binding assay using the Monolith NT.115 MST device (NanoTemper technologies). His-tagged SfMCT and SfMCT-R280A were affinity-purified in *n*-dodecyl β-D-maltopyranoside (DDM, Glycon Biochemicals GmbH) as described in the Purification section. The protein was eluted from the Ni-NTA column using an imidazole-containing buffer (20 mM Bis-Tris propane-HCl (pH 8), 150 mM NaCl, 400 mM imidazole, 0.03% (w/v) DDM) followed by buffer exchange into MST buffer (20 mM Bis-Tris propane-HCl (pH 8), 150 mM NaCl, 0.03% (w/v) DDM) using a desalting column (Zeba spin desalting columns 7k MWCO, Thermo Scientific). Proteins were labelled using the His-tag labeling kit (NanoTemper technologies) as described by the manufacturer with minor mod-ifications. Instead of using phosphate-buffered saline supplemented with Tween 20 for the dilution of the protein and the labeling dye, MST buffer was used. A binding experiment consisted of 16 samples (20 μl), which contained 50 nM labeled SfMCT or SfMCT-R280A, and L-lactate at a concentration from 7.6 μM to 250 mM. A 500 mM stock solution of L-lactate was prepared in MST buffer. The samples were loaded into coated glass capillaries (Premium Capillaries, NanoTemper technolo-gies). The MST measurements were performed under ambient conditions using medium MST power and 20% LED excitation power. The $K_d$ value was calculated using the mass action equation (NanoTemper technologies). The average $K_d$ value was determined from 5 independent experiments.

**Sequence alignment and sequence logo generation.** Amino acid sequence alignment was performed with Clustal Omega[48]. The UniProt ID codes of SfMCT, human MCT1, human MCT2, human MCT3 and human MCT4 are A0LNN5, P53985, O60669, O95907 and O15427, respectively. The sequence logo repre-sentation of the region close (+/−11 amino acids) to N276 of SfMCT was com-puted using a multiple sequence alignment of SfMCT and bacterial homologues. Bacterial homologues of SfMCT were identified by searching the bacterial target database of UniProt Knowledgebase on ExPASy using the BLAST algorithm (https://web.expasy.org/blast/) with the SfMCT amino acid sequence (UniProt ID A0LNN5) as template. The first 50 sequences were selected and aligned with Clustal Omega using the BLOSUM 62 substitution matrix[48]. The sequence logo was computed using WebLogo[49].

**Reporting summary.** Further information on research design is available in the Nature Research Reporting Summary linked to this article.

## Data availability

Atomic coordinates for the TSA and L-lactate bound crystal structures of SfMCT have been deposited in the Protein Data Bank under accession numbers 6G9X (TSA) and 6HCL (L-lactate). The source data underlying Figs 1a, b, 3d, 5d, 6c–e, Supplementary Figs 1, 6 and 7 are provided as a Source Data file. Other data are available from the corresponding author upon reasonable request.

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

## Acknowledgements

We thank Laura Baldomà (University of Barcelona, Spain) for providing the *E. coli* JA202 strain and the staff of the SLS (Paul Scherrer Institute) X06SA beamline for excellent support and advice. Financial support from the University of Bern, the Swiss National Science Foundation (grant 31003A_162581) and the NCCR TransCure is kindly acknowledged.

## Author contributions

P.D.B., D.K., S.B. and D.F. designed the experiments, analyzed the data and wrote the manuscript. All authors discussed the experiments, read and approved the manuscript.

## Additional information

**Competing Interests:** The authors declare no competing interests.

**Peer Review Information:** *Nature Communications* thanks Ulrich Schweizer and other anonymous reviewer(s) for their contribution to the peer review of this work. Peer reviewer reports are available.

