## [Peer Review File · Nature Communications]

Reviewers' comments:

Reviewer #1 (Remarks to the Author):

In this work from Fotiadis lab authors describe the crystal structures of bacterial homologue of human MCT1/4 transporters, belonging to solute carrier family 16 of transporters. Although these structures show the well studied MFS fold, nevertheless this structural information is an essential piece to understand the working mechanism of lactate transport. The work is technically sound and in general well described. However I suggest authors to look at a few more points related to the transport and binding to make this manuscript even clearer. The crucial experiment will be to show (in) dependence of transport on electrochemical potential and sodium gradient and not only proton. The authors should also clarify the role of protons - are they transported (and it sounds as it is implied from discussion, lines 305-313) or they just stimulate the transport? If former is the case, the authors should discuss the symport of lactate and protons and confirm the intracellular acidification (by using proton specific dyes - pyranine or ACMA, there are several examples in the literature - Ehrnstorfer et al., 2017, Nature Communications 8: 14033; Gati et al., 2017, Nature Communications 8: 1313; Prabhala et al., 2014, FEBS letters 588(4):560-5; etc.)

For lactate binding it would be beneficial to measure K_d values with ITC and not just infer those from kinetics. Is there any cooperativity of binding two lactate molecules?

F335A mutation sounds intriguing - can the authors show that indeed it allows for transport of bulkier substrates using the transport assay they established?

At some points the language can be improved - e.g. line 173 - 'This releases' - this removes ; line 191 - 'agreeing with' - in line with;

Line 215 - 'alternatingly' - in alternative fashion

Line 315 'in the presented structures and resolutions' in the presented structures at given resolutions.

Typo in line 853 NH_4SO_4 should be $(\text{NH}_4)_2\text{SO}_4$

I think the authors can state in the manuscript the resolutions for structures they used for refinement, i.e. 2.30Å and 2.50Å instead of 2.54Å and 2.69Å with the notion that it is anisotropic, but both ways are fine.

Reviewer #2 (Remarks to the Author):

In the present manuscript, the authors describe the crystal structure of bMCT in the outward-open conformation, a protein homologous to members of the solute carrier family 16 (MCT). bMCT shares amino acid identity with MCT1 (SLC16A1) and MCT4 (SLC16A3) which are both transporters for lactate. The authors were able to identify several amino acids involved in substrate binding and pH sensing. Expression of MCT1 and MCT4 play a pivotal role for cancer survival and therefore the transporters are interesting targets for structure-based drug design. Crystal structures of any MCT are not available at the moment and existing homology models are all based on the bacterial glycerol-3-phosphate transporter or sugar transporters.

Comments

While the authors mention the published MCT1/4 homology models, they do not discuss them in detail. This would be important in order to understand whether the new bMCT structure is a real improvement for the field. So are the helices and loops correctly assigned? Does the overall topology fit? Are active site residues at predicted sites? When MCT1 was initially cloned, it was a point mutant that altered its substrate specificity. In MCT8/MCT10 the corresponding amino acid is involved in substrate specificity. Is the corresponding amino acid at a strategic position according to the bMCT crystal structure? The whole paper is written more from a bacterial perspective, but the relevance and impact come from human biology. The field would profit more from putting the structure into context with what is known about eukaryotic MCTs.

MCT1 and MCT4 need basigin as accessory protein which is necessary for the translocation of the proteins and their function. Most likely, the expression of bMCT in prokaryotes does not depend on such proteins. This should be discussed, because that would mean that the accessory subunits are not playing a role in transport function. Would the independence from accessory proteins argue

that bMCT is not a “real” MCT? Are there MCTs that do not require accessory subunits? Is anything known about their role in biology?

MCT1 and MCT4 are known to transport substrates in both directions. However, depending on the tissue the preference of MCT1 is the import of lactate while MCT4 preferentially exports lactate. Is bMCT also capable of lactate export?

The authors demonstrate that Arg280 (and Asn276) play a pivotal role in substrate binding. The corresponding amino acids in ratMCT1 are Asp302 and Arg306 which are reported to be involved in the binding of the MCT1-specific (but not MCT4-specific) inhibitor AR-C155858. Does AR-C155858 also inhibit bMCT-mediated lactate transport?

The authors performed functional studies using different bMCT mutants. Western blot analyses show different expression levels of the mutant proteins. The authors should show that equal amounts of protein were loaded on the gel using a bacterial protein for normalization. In addition, some of the mutants show lower expression levels than others compared to WT bMCT. How can the authors be sure that a decrease in transport activity is an effect of disturbed protein function and not the result of low protein expression?

I do not have any methodological concerns.

Reviewer #3 (Remarks to the Author):

Bosshart et al present the first structures of a bacterial SLC16 member with bound L-lactate and TSA, respectively. Both are found in an outward-facing conformation. The substrate binding site and a potential proton sensor are characterized and discussed.

The structures as far as I can judge from the given parameters and the figures are of high quality and well assigned. However, I have several problems with the manuscript that need to be solved before its acceptance:

1) Just a small comment about the structure: The phasing was done with a data set of max 2.58 Å while the final resolution was 2.54 Å. I assume this was achieved by phase extension using the high-resolution dataset, however, this is neither mentioned in the result nor in the methods section.

2) The introduction completely focuses on the human homologs instead of properly introducing the family and its general features. This way it is very difficult to follow some of the aspects discussed in the results part as they were not properly introduced (e.g. overall topology, proton sensor). Generally spoken I do not agree with selling these structures as such good models for MCT1 and MCT4 since a formal proof is missing. In line with this, a sequence identity of 25% is definitely not to be considered high. Nonetheless, it is of course relevant to solve any first structure of the SLC16 family just because we learn a lot about its general architecture!

3) Calling the protein just bMCT is rather uncommon; you have shown by an alignment that there are different bacterial SLC16 members. Instead, as known for other bacterial homologs, I suggest SfMCT.

4) The uptake assay:

- The authors continuously claim that they are observing inhibition by the other substrates tested, which would mean that those are not transported. However, all tested substrates that led to a reduced uptake of L-lactate could as well be transported. What you observed would then be simple competition for transport instead of just binding. To investigate whether the other substrates are actually transported a counterflow or exchange assay on prepared membranes or even better proteoliposomes should be performed. If then in fact the other substrates are hardly transported, the authors can claim that bMCT is highly specific for lactate.

- Also a time course of the transport is required to better judge the obtained data.
- Since TSA was bound in the crystal structure it should also be included in the competition assay in the initial characterization.
- Just because CCCP inhibits transport in whole cells, H⁺ coupling cannot be concluded. The authors should include the pH dependency of transport already in the initial characterization.
- I miss the control measurements confirming the missing uptake of the none-complemented strain.

5) Overall structure: The extensive description of the crystal packing is rather useless. Instead, the authors should introduce the overall architecture of the protein in much more detail. Accordingly, proper figures highlighting the overall structure, the pseudo-twofold symmetry, the binding pocket,... should be provided, while main figure on the crystal packing should be removed.

6) Substrate and ligand binding pocket:

- First of all it is not clear to me, why you distinguish substrate and ligand binding pocket. What do you consider a ligand? TSA? For the above given reason I doubt your conclusion that TSA only binds, which also questions the definition of a K_i-value for TSA.
- In general, this whole section is rather difficult to follow and has to be better structured. I suggest to first focus at the likely more relevant binding of lactate and then shortly discuss the TSA binding.
- Additionally, it would be very helpful to add a figure, which localizes the binding site within the overall structure.
- How high is the sequence identity of the binding site with the binding sites in MCT1 and MCT4. If the sequence identity with respect to the binding site is high, a supplementary figure modeling those binding sites in comparison to the given binding site would be very helpful to follow the authors' general argumentation.

7) Proton sensor:

- In general, as for the paragraph above, I find it difficult to follow the argumentation. E.g. why are you suddenly referring to lactate binding again (line 230)?
- To be honest, I doubt the existence of the proton sensor and I miss a proper introduction into the concept of this sensor. What would be the physiological relevance for a bacterium? Isn't the proton sensor rather important for the human homologs, which seem to fulfill different jobs probably even in the same tissue? The histidine is not even conserved... Instead, I hypothesize that the residues that show the strongest effect upon mutating (no transport) are actually involved in proton binding for its co-transport. Perhaps the mutations causing increased uptake at alkaline conditions actually changed the coupling of lactate to protons transported...
- A figure that localizes the discussed 'sensor' area within the overall structure would be helpful.
- If the authors disagree with my hypothesis, where do you expect protonation/deprotonation for the co-transport? I am missing a clear separation of both aspects.
- What are the data in figure 6C normalized to? Why start the wildtype and K377A not at 100%

8) The authors claim that they "provide a framework for understanding the molecular working mechanisms of substrate binding and transport of SLC16 transporters" (lines 254f). I disagree to this statement. They only described the binding site for substrate while neither discussing about the coupling nor the induced conformational changes, which is required for a transport MECHANISM.

Point-by-point discussion to Bosshart *et al.* NCOMMS-18-34613A

General comments to the Reviewers:

- For better clarity and to improve the understandability of the presented functional data, we have included a new Supplementary Figure (Supplementary Figure 2) containing the molecular structures of the physiologically relevant competitors used to determine the substrate specificity of the transporter.
- We have uploaded a PDF of the manuscript where all changes were tracked and are displayed.

Reviewer #1 (Remarks to the Author):

In this work from Fotiadis lab authors describe the crystal structures of bacterial homologue of human MCT1/4 transporters, belonging to solute carrier family 16 of transporters. Although these structures show the well studied MFS fold, nevertheless this structural information is an essential piece to understand the working mechanism of lactate transport. The work is technically sound and in general well described.

However I suggest authors to look at a few more points related to the transport and binding to make this manuscript even clearer.

The crucial experiment will be to show (in) dependence of transport on electrochemical potential and sodium gradient and not only proton.

Authors:

The presented uptake experiments were performed in a buffer, which only contained potassium and no sodium. [¹⁴C]L-lactate was efficiently transported indicating that the function of SfMCT* is sodium-independent. Regarding the electrochemical potential, we measured the uptake of [¹⁴C]L-lactate in the presence of valinomycin, which had no effect on L-lactate transport. Please see Supplementary Fig. 1c.

* As suggested by Reviewer 3, we have changed the name from bMCT to SfMCT (Sf for *Syntrophobacter fumaroxidans*) in the revised version of our manuscript.

The authors should also clarify the role of protons - are they transported (and it sounds as it is implied from discussion, lines 305-313) or they just stimulate the transport? If former is the case, the authors should discuss the symport of lactate and protons and confirm the intracellular acidification. (by using proton specific dyes - pyranine or ACMA, there are several examples in the literature - Ehrnstorfer *et al.*, 2017, Nature Communications 8: 14033; Gati *et al.*, 2017, Nature Communications 8: 1313; Prabhala *et al.*, 2014, FEBS letters 588(4):560-5; etc.)

Authors:

We thank the Reviewer for raising this good point that we have addressed with a similar approach as described in the suggested reference Prabhala *et al.*, 2014, FEBS Letters 588(4):560-5, i.e., by measuring the change of the extracellular pH associated with proton-coupled substrate transport. Using a micro pH electrode* in a solution containing SfMCT expressing bacteria, we monitored changes in the extracellular pH. Addition of L-lactate to such a suspension resulted in a net pH increase, i.e., removal of protons from the extracellular solution. This observed increase in pH reflects the SfMCT mediated co-transport of protons together with L-lactate from the extracellular space into bacteria. We have added these results in Supplementary Fig. 1e.

* Such a set-up was also used to measure proton-coupled sugar transport through LacY (Franco and Brooker, 1994, J. Biol. Chem. 269(10):7379-86).

For lactate binding it would be beneficial to measure K_d values with ITC and not just infer those from kinetics. Is there any cooperativity of binding two lactate molecules ?

Authors:

We agree with the Reviewer that gaining insight into L-lactate binding would be interesting. Therefore, we used microscale thermophoresis (MST) and obtained a K_d value of ~3 mM for L-lactate binding to purified, detergent-solubilized SfMCT. As a negative control we also measured the L-lactate binding affinity of the transport-deficient mutant SfMCT-R280A by MST. As expected, SfMCT-R280A did not show L-lactate binding. The corresponding results were added to the revised manuscript, please see Supplementary Fig. 6a,b.

To our knowledge such a relatively high K_d value of ~3 mM strongly limits the reliable usability of ITC to measure substrate affinity and particularly to determine the binding stoichiometry between SfMCT and L-lactate. The latter

is also the case for MST, which at least allows K_d determination using reasonable amounts of purified membrane protein.

We are aware of the discrepancy between the kinetic K_m -value, which is in the upper μM -range, and the K_d -value, which is in the lower mM -range. It is important to highlight that the K_d -value was determined from detergent-solubilized, purified protein. This state of SfMCT might not have the same substrate affinity properties than SfMCT that is embedded in a lipid bilayer. Such discrepancies have already been observed for other proton-coupled MFS transporters such as the peptide transporter PepT_{St} ($K_{m,\text{app}} = 30 \mu\text{M}$, $K_d = 8.59 \text{ mM}$ (MST measurement); Solcan et al., 2012, EMBO J. 31(16):3411-21 and Martinez Molledo et al., 2018, Structure, 26(3):467-76). Therefore, we consider the provided K_m -value as the most important functional parameter.

F335A mutation sounds intriguing - can the authors show that indeed it allows for transport of bulkier substrates using the transport assay they established ?

Authors:

Indeed, F335A is an intriguing and interesting mutant. To address this question, we tested a slightly larger physiologically relevant monocarboxylate, i.e., acetoacetate, to compete with $[^{14}\text{C}]\text{L}$ -lactate transport through SfMCT-F335A overexpressed in bacteria. Acetoacetate revealed a significantly stronger reduction of the radioligand uptake through SfMCT-F335A compared to WT bMCT, i.e., residual $[^{14}\text{C}]\text{L}$ -lactate uptake F335A: 43% and WT: 86%. Therefore, we conclude that F335 is indeed involved in tuning of the substrate specificity. We have added this result in Supplementary Fig. 1g.

At some points the language can be improved - e.g.

line 173 - 'This releases' - this removes

Authors: We included this suggestion in the revised manuscript.

Line 191 - 'agreeing with' - in line with

Authors: We included this suggestion in the revised manuscript.

Line 215 - 'alternatingly' - in alternative fashion

Authors: We replaced "alternatingly" by "in an alternating fashion".

Line 315 'in the presented structures and resolutions' - in the presented structures at given resolutions

Authors: We included this suggestion in the revised manuscript.

Typo in line 853 NH_4SO_4 should be $(\text{NH}_4)_2\text{SO}_4$

Authors: Thank you for spotting this - we corrected the typo in the revised manuscript.

I think the authors can state in the manuscript the resolutions for structures they used for refinement, i.e. 2.30\AA and 2.50\AA instead of 2.54\AA and 2.69\AA with the notion that it is anisotropic, but both ways are fine.

Authors:

Thank you for this comment. We discussed these two possibilities and prefer to keep the original (conservative) numbers.

Reviewer #2 (Remarks to the Author):

In the present manuscript, the authors describe the crystal structure of bMCT in the outward-open conformation, a protein homologous to members of the solute carrier family 16 (MCT). bMCT shares amino acid identity with MCT1 (SLC16A1) and MCT4 (SLC16A3) which are both transporters for lactate. The authors were able to identify several amino acids involved in substrate binding and pH sensing. Expression of MCT1 and MCT4 play a pivotal role for cancer survival and therefore the transporters are interesting targets for structure-based drug design. Crystal structures of any MCT are not available at the moment and existing homology models are all based on the bacterial glycerol-3-phosphate transporter or sugar transporters.

Comments

While the authors mention the published MCT1/4 homology models, they do not discuss them in detail. This would be important in order to understand whether the new bMCT structure is a real improvement for the field.

Authors:

We agree with the Reviewer that a comparison of our experimental structures with the *in silico* homology models of rat MCT1 and MCT4, and their discussion is important and necessary. Therefore, we have added this discussion (first paragraph of Discussion) in the revised version of the manuscript.

In our opinion the here provided experimental structures of an SLC16 L-lactate transporter represent a major contribution to the field since the published homology models need to be significantly improved – please see our answer to the next question.

So are the helices and loops correctly assigned ? Does the overall topology fit ?

Authors:

Homology models of rat MCT1 and MCT4 have been previously published (Wilson et al., 2005, J. Biol. Chem. 280:27213-21; Manoharan et al., 2006, Mol. Membr. Biol. 23:486-98; Wilson et al., 2009, J. Biol. Chem. 284:20011-21). These models are based on the inward-open structure of the glycerol-3-phosphate:phosphate antiporter GlpT, which belongs to a completely different MFS subfamily (i.e., the organophosphate:phosphate antiporter family) and shares low sequence identity with rat MCT1 and MCT4. Importantly, functionally relevant residues are not conserved between GlpT and MCTs. It should be noted that the outward-open MCT1 model was generated by manipulating the available inward-open model (Wilson et al., 2009, J. Biol. Chem. 284:20011-21), which is problematic. Thus, this model was not computed using a template in the pharmacologically relevant outward-open conformation.

As requested, we analyzed the secondary structure/topology of this outward-open MCT1 homology model with our outward-open SfMCT model. For that purpose we used the 2struc web interface (<http://2struc.cryst.bbk.ac.uk>; Klose et al., 2010, Bioinformatics 26:2624-25) where the DSSP (Kabsch and Sander, 1983, Biopolymers 22:2577-637) and Stride (Frishman and Argos, 1995, Proteins 23:566-79) algorithms were applied to determine the secondary structure elements. Many transmembrane spanning domains of the rat MCT1 homology model were fragmented consisting of relatively short alpha-helices, and a considerable amount of hydrogen-bonded turns, bends and 3_{10} -helices (please see Supplementary Fig. 11a). Importantly, transmembrane spanning domain 2 is distorted and no alpha-helical structure was detected at all. Surprised by this result, we analyzed all available X-ray structures of MFS transporters using the above-mentioned algorithms and concluded that such a high fraction of non-alpha-helical structural elements is not observed in transmembrane domains of prokaryotic and eukaryotic MFS transporters.

We checked the validity of the topology of the rat MCT1 homology model by overlaying it with our experimental SfMCT* structure and obtained a high r.m.s.d. of 6.44 Å (over 1882 atoms). In summary, this structural alignment shows that the topologies of the rat MCT1 homology model and the experimental X-ray structure of SfMCT are roughly comparable, but the secondary structure is significantly different (please see Supplementary Fig. 11a).

In summary and based on the above mentioned points, we consider that the SfMCT structure represents a real and major improvement for the field.

* As suggested by Reviewer 3, we have changed the name from bMCT to SfMCT (Sf for *Syntrophobacter fumaroxidans*) in the revised version of our manuscript.

Are active site residues at predicted sites?

Authors:

We have defined the active (ligand binding) site of SfMCT as residues that are located within a distance of 4 Å from the bound thiosalicylate and/or L-lactate molecule(s), which were clearly identified in the presented structures of SfMCT. Accordingly, the highly conserved arginine (e.g., SfMCT: R280; human MCT1: R313; rat MCT1: R306; human MCT4: R278; rat MCT4: R282), which is crucial for L-lactate binding (and transport) in SfMCT and mammalian MCTs, is located in TM8 in line with the homology models.

However, alignment of the SfMCT structure with the model of rat MCT1 uncovered significant differences between the locations and orientations of functional key residues - please see Supplementary Fig. 11b for details.

In summary, incorrect and unprecise locations/orientations of functional key residues in the rat MCT1 homology model preclude the reliable use of structure-based applications, e.g., molecular dynamics simulations and structure-based drug design.

When MCT1 was initially cloned, it was a point mutant that altered its substrate specificity. In MCT8/MCT10 the corresponding amino acid is involved in substrate specificity. Is the corresponding amino acid at a strategic position according to the bMCT crystal structure?

Authors: Indeed, when MCT1 was initially cloned it was recognized that the gene, which codes for a mevalonate transporter, differed from the wild-type version by a phenylalanine to cysteine exchange (human MCT1: F367C) in the predicted TM10 (Kim et al., 1992, J. Biol. Chem. 267:23113-21). However, the wild-type version of this transporter does not transport mevalonate.

The respective phenylalanine residue corresponds to Y331 in SfMCT, which is also located in TM10 (see Supplementary Fig. 9). It faces towards the substrate translocation path (please see Fig. 3b,c). According to our ligand-bound structures, it belongs to the ligand binding site and we have shown that an aromatic side chain (i.e., tyrosine or phenylalanine) at this position is essential for L-lactate transport (please see Fig. 5d). Furthermore, we demonstrated that the hydroxyl-group of Y331 is important for the recognition of thiosalicylate (TSA). The conservative replacement Y331F reduced SfMCT's affinity for TSA almost tenfold (please see Fig. 3d). Therefore, the mentioned amino acid residue is indeed located at a strategic position according to the SfMCT crystal structure and it is involved in substrate specificity of SfMCT.

In the revised version of our manuscript we have mentioned and discussed the phenylalanine to cysteine mutant in the context of our structure and its key residue Y331 (second paragraph of Discussion).

The whole paper is written more from a bacterial perspective, but the relevance and impact come from human biology. The field would profit more from putting the structure into context with what is known about eukaryotic MCTs.

Authors: We thank the Reviewer for this comment. In the revised manuscript we expanded the discussion of the functionally important and conserved arginine that is located in TM8 (SfMCT: R280; human MCT1: R313; rat MCT1: R306; human MCT4: R278; rat MCT4: R282). It has been shown that this residue is essential for transport in many eukaryotic MCTs and cannot even be replaced conservatively (i.e., arginine to lysine). It has been shown that the exchange of this critical arginine residue to a glutamine in human MCT1 can be found in patients suffering from severe ketoacidosis (van Hasselt et al., 2014, N. Engl. J. Med. 371:1900-7). The resulting non-functional MCT1 is not capable to bind and transport monocarboxylic ketone bodies into cells.

Our structures with bound L-lactate and TSA provide the structural explanation for the critical location and role of this arginine residue.

In addition to the above discussed arginine residue, we have expanded the discussion on the role of the functionally important aromatic residue in TM10 (SfMCT: Y331; human MCT1: F367; human MCT4: Y332), which seems to be involved in substrate specificity.

MCT1 and MCT4 need basigin as accessory protein which is necessary for the translocation of the proteins and their function. Most likely, the expression of bMCT in prokaryotes does not depend on such proteins. This should be discussed, because that would mean that the accessory subunits are not playing a role in transport function.

Authors: We thank the Reviewer for raising this important point. Basigin (CD147) is needed for proper plasma membrane expression of MCT1 and MCT4 in eukaryotic cells (Kirk et al., 2000, EMBO J. 19:3896-904). However, it was recently shown that the MCT4-basigin heterodimer can be disrupted in the plasma membrane without effect on L-lactate transport through MCT4 (Voss et al., 2017, Sci. Rep. 7, 4292). This highlights that basigin is not essential for the transport function of MCT4.

In the genome of *E. coli*, which was used for the SfMCT transport studies, there is no basigin homologue. The fact that L-lactate was efficiently transported through the SLC16 homologue SfMCT shows that surface expression and function of SfMCT do not depend on a bacterial CD147 homologue.

In the revised version, we have introduced the ancillary proteins basigin and embigin (first paragraph Introduction).

Would the independence from accessory proteins argue that bMCT is not a "real" MCT? Are there MCTs that do not require accessory subunits?

Authors: Our data, for example amino acid sequence, substrate profile, conservation of key residues (e.g., R280 and Y331) and proton-dependence suggest that SfMCT is a prokaryotic SLC16 family member. As discussed in the previous point, association of a SLC16 transporter with an accessory protein is not a prerequisite for function (Voss et al., 2017, Sci. Rep. 7, 4292). Similar to the L-type amino acid transporters (LATs) from the SLC7 family that are trafficked to the plasma membrane by association with an ancillary protein (e.g., CD98) in higher organisms, some MCTs also need ancillary proteins for correct trafficking (Fotiadis et al., 2013, Mol. Aspects Med.

34, 139-58). As for SfmMCT, prokaryotic SLC7 amino acid transporters are correctly expressed and fully functional in bacteria (e.g., the L-arginine-aggmatine transporter AdiC; Iyer et al. 2003, J. Bacteriol. 185, 6556-61). Furthermore, it has been shown that the SLC16 family member MCT8 does not form heterodimers with ancillary proteins (Visser et al., 2009, Endocrinology 150, 5163-70). Therefore, association of MCTs with an ancillary protein is not a prerequisite for all family members and SfmMCT can be regarded as a "real" MCT.

Is anything known about their role in biology?

Authors: Basigin (CD147) is expressed in different cell types such as epithelial and endothelial cells but also in immune cells. Glycosylation is important for trafficking of many membrane proteins to the plasma membrane of eukaryotic cells. Certain membrane proteins (e.g., MCT1 and MCT4), which are not glycosylated, require the association with glycosylated ancillary proteins such as basigin (CD147) for correct and efficient trafficking. It has been recently shown that CD147 is involved in the interaction between MCTs and a membrane-anchored carbonic anhydrase, which catalyzes the reversible hydration of CO₂ to HCO₃⁻ and H⁺ thus facilitating the transport activity of the MCTs (Forero-Quintero et al., 2019, J. Biol. Chem, 294:593-607). Furthermore, basigin is involved in the release/activation of matrix metalloproteinases, it promotes angiogenesis via the secretion of the vascular endothelial growth factor, it is involved in the migration and metastasis of tumor cells, it regulates the chemoresistance of certain tumor cells and basigin regulates the ABC-transporter-driven transport of methotrexate in immune-cells (reviewed in Xiong et al., 2014, Int. J. Mol. Sci. 15:17411-41).

Since the focus of our manuscript is not on ancillary proteins such as basigin, we have added a brief comment regarding the importance of accessory proteins for correct MCT trafficking to the plasma membrane in the revised version of our manuscript (first paragraph Introduction).

MCT1 and MCT4 are known to transport substrates in both directions. However, depending on the tissue the preference of MCT1 is the import of lactate while MCT4 preferentially exports lactate. Is bMCT also capable of lactate export?

Authors: Despite extensive efforts we were not able to establish an efflux assay for SfmMCT to test this hypothesis. However, MCT1 and MCT4 have K_m -values of 3-5 mM and 20-35 mM, and seem therefore to be "tuned" for L-lactate uptake and export, respectively. Based on the K_m s of MCT1 and MCT4, and the here determined K_m of 233 μ M of SfmMCT, it is most probable that SfmMCT works as a L-lactate importer in *Syntrophobacter fumaroxidans*. It should also be mentioned that most secondary active transporters are able to transport in both directions depending on the electrochemical potential difference. Therefore, SfmMCT should also be able to export under certain conditions of the electrochemical potential.

The authors demonstrate that Arg280 (and Asn276) play a pivotal role in substrate binding. The corresponding amino acids in ratMCT1 are Asp302 and Arg306 which are reported to be involved in the binding of the MCT1-specific (but not MCT4-specific) inhibitor AR-C155858. Does AR-C155858 also inhibit bMCT-mediated lactate transport?

Authors: Thank you for this comment. We have performed a competition experiment where we measured the uptake of [¹⁴C]L-lactate into SfmMCT expressing bacteria in the presence of 100 μ M AR-C155858. Although the used inhibitor concentration was significantly above the reported K_i -value of ~2 nM (Ovens et al., 2010, Biochem. J. 425:523-30), no reduction of [¹⁴C]L-lactate uptake was observed (control: 100%, 100 μ M AR-C155858: 103%). Therefore, we conclude that this MCT1-specific inhibitor does not bind to SfmMCT and does not act as a transport inhibitor in our case. We prefer not to add this data to the revised manuscript since we are convinced that this information does not provide an added value and might be confusing to the reader.

The authors performed functional studies using different bMCT mutants. Western blot analyses show different expression levels of the mutant proteins. The authors should show that equal amounts of protein were loaded on the gel using a bacterial protein for normalization.

Authors: Thank you for raising this point, which we addressed in the revised manuscript. To our knowledge there is no bacterial protein such as actin in mammalian cells, which is routinely used as loading control. Therefore, we provide new Western blots together with Coomassie-stained SDS-PAGE gels of the loaded, lysed bacteria (Supplementary Fig. 7). These stained gels demonstrate that in all cases similar amounts of lysed bacteria were loaded.

In addition, some of the mutants show lower expression levels than others compared to WT bMCT. How can the authors be sure that a decrease in transport activity is an effect of disturbed protein function and not the result of low protein expression ?

Authors: Indeed, this is a general problem in the transporter field. However, the concerned minority of mutants are basically completely inactive, but still indicate significant expression levels. Therefore, we are convinced that the drawn conclusions are valid.

I do not have any methodological concerns.

Authors: Thank you. We appreciate the provided feedback that in our opinion strengthened the manuscript.

Reviewer #3 (Remarks to the Author):

Bosshart et al present the first structures of a bacterial SLC16 member with bound L-lactate and TSA, respectively. Both are found in an outward-facing conformation. The substrate binding site and a potential proton sensor are characterized and discussed.

The structures as far as I can judge from the given parameters and the figures are of high quality and well assigned. However, I have several problems with the manuscript that need to be solved before its acceptance:

1) Just a small comment about the structure: The phasing was done with a data set of max 2.58 Å while the final resolution was 2.54 Å. I assume this was achieved by phase extension using the high-resolution dataset, however, this is neither mentioned in the result nor in the methods section.

Authors: We thank the Reviewer for this comment: In “Methods” of the revised manuscript we have included additional information regarding phasing.

2) The introduction completely focuses on the human homologs instead of properly introducing the family and its general features. This way it is very difficult to follow some of the aspects discussed in the results part as they were not properly introduced (e.g. overall topology, proton sensor).

Authors: Indeed, the introduction was suboptimal. In the revised version we have extended the Introduction section and added relevant information about the SLC16 family (e.g., expression pattern, topology, proton dependence and more) for better understanding of the presented results (please see revised Introduction).

Generally spoken I do not agree with selling these structures as such good models for MCT1 and MCT4 since a formal proof is missing. In line with this, a sequence identity of 25% is definitely not to be considered high. Nonetheless, it is of course relevant to solve any first structure of the SLC16 family just because we learn a lot about its general architecture!

Authors: We appreciate this critical feedback. In the revised version, we have changed the expression “high sequence identity” into “significant sequence identity” (last paragraph in Introduction).

The multiple sequence alignment that we provided in the Supplementary Information (Supplementary Fig. 9) shows that key residues are identical or similar in SfMCT*, and MCT1-4. Furthermore, in the SLC16 family only MCT1-4, and MCT8 and MCT10 are well-characterized. While MCT1-4 are proton-driven monocarboxylate transporters, MCT8 and MCT10 encode high-affinity thyroid hormone and aromatic amino acid transporters, respectively (Visser et al., 2011, Mol. Endocrinol. 25:1-14). These two transporters are facilitators, in contrast to MCT1-4 and SfMCT, which are proton-coupled transporters (shown for SfMCT in this work). Importantly, we have demonstrated that SfMCT has no affinity for amino acids (Fig. 1b). In summary and because of these common features, we are convinced that the structure of SfMCT, which is the first structure of the SLC16 family, is a valid model for MCT1-4.

* As you suggested (please see below), we have changed the name from bMCT to SfMCT (Sf for *Syntrophobacter fumaroxidans*) in the revised version of our manuscript.

3) Calling the protein just bMCT is rather uncommon; you have shown by an alignment that there are different bacterial SLC16 members. Instead, as known for other bacterial homologs, I suggest SfMCT.

Authors: We thank the Reviewer for this valuable suggestion that we have adopted.

4) The uptake assay:

- The authors continuously claim that they are observing inhibition by the other substrates tested, which would mean that those are not transported. However, all tested substrates that led to a reduced uptake of L-lactate could as well be transported. What you observed would then be simple competition for transport instead of just binding. To investigate whether the other substrates are actually transported a counterflow or exchange assay on prepared membranes or even better proteoliposomes should be performed. If then in fact the other substrates are hardly transported, the authors can claim that bMCT is highly specific for lactate.

Authors: We agree with the Reviewer that using the word “inhibition” in the context of the presented assay was not appropriate. The assay, which we used to determine the substrate specificity, is based on the competition between unlabelled molecules and radioactive [¹⁴C]L-lactate. Therefore, we replaced the word “inhibition” by “competition” at the corresponding places in the revised manuscript. We thank for this valuable feedback, which might have raised confusions.

We also agree with the Reviewer that molecules that reduced the transport of [¹⁴C]L-lactate through SfMCT can be either transported or act as (non-transported) inhibitors. To distinguish between the two possibilities, we measured SfMCT-mediated transport of radioactively labelled physiologically relevant monocarboxylates, which reduced the uptake of [¹⁴C]L-lactate into bacteria by at least 40 % (i.e., propionate, butyrate and pyruvate – Fig. 1b). According to these experiments, propionate, butyrate and pyruvate are not transported by SfMCT. We have added these results in the revised manuscript (Supplementary Fig. 1b). Based on these results, we suggest that SfMCT is highly specific for L-lactate.

- Also a time course of the transport is required to better judge the obtained data.

Authors: We have provided a graph showing the results of a time course experiment of the transport in Supplementary Information (Supplementary Fig. 1a).

- Since TSA was bound in the crystal structure it should also be included in the competition assay in the initial characterization.

Authors: As requested, we have added the competition data of TSA in the graph of the competition assay (Fig. 1b) in the revised manuscript.

- Just because CCCP inhibits transport in whole cells, H⁺ coupling cannot be concluded. The authors should include the pH dependency of transport already in the initial characterization.

Authors: We have provided a graph showing the pH-dependent uptake of [¹⁴C]L-lactate in Supplementary Fig. 1d.

- I miss the control measurements confirming the missing uptake of the none-complemented strain.

Authors: We would like to mention that [¹⁴C]L-lactate uptake by *E. coli* JA202 (MC4100 glcA::cat lldP::kan) that are transformed with an empty plasmid is not completely abrogated. There is a low basal uptake level, which is four to five times lower than the uptake into SfMCT-overexpressing bacteria. We have added the corresponding data in Supplementary Fig. 1a.

5) Overall structure: The extensive description of the crystal packing is rather useless. Instead, the authors should introduce the overall architecture of the protein in much more detail. Accordingly, proper figures highlighting the overall structure, the pseudo-twofold symmetry, the binding pocket,... should be provided, while main figure on the crystal packing should be removed.

Authors: We agree with the Reviewer that the manuscript does not significantly benefit from the crystal contact description in the main text. Therefore, we transferred this information to Supplementary Information

(Supplementary Fig. 4 and corresponding legend). As requested, we describe the overall architecture of SfmMCT in more detail in the main manuscript (please see section “Overall structure of SfmMCT” in Results). Furthermore, we added an additional panel to Fig. 3, Fig. 5 and Fig. 6, which show the locations of the binding pocket and the proton binding site in the SfmMCT protein. We feel that the introduction of these panels based on the Reviewer’s suggestion will help better understanding the SfmMCT structure.

6) Substrate and ligand binding pocket:

- First of all it is not clear to me, why you distinguish substrate and ligand binding pocket. What do you consider a ligand? TSA? For the above given reason I doubt your conclusion that TSA only binds, which also questions the definition of a K_i -value for TSA.

Authors: Based on our functional results, L-lactate is transported by SfmMCT and is therefore a substrate. On the other hand, we consider a molecule, which interacts with SfmMCT and has an effect on the transport of [^{14}C]L-lactate as a ligand. To avoid confusion, we use now the term ligand appropriately in the text, which is correct for a transported (substrate) and a non-transported molecule.

To investigate if TSA is transported by SfmMCT, we used a micro pH electrode-based assay that was used in the past to measure proton-coupled sugar transport through LacY (Franco and Brooker, 1994, J. Biol. Chem. 269:7379-86). Using a micro pH electrode in a solution containing SfmMCT expressing bacteria, the change in the extracellular pH was monitored. Addition of L-lactate to such a suspension resulted in a net pH increase, i.e., removal of protons from the extracellular solution (Supplementary Fig. 1e). This observed increase in pH reflects the SfmMCT-mediated co-transport of protons together with L-lactate into bacteria. However, addition of TSA did not lead to a pH increase indicating that TSA is not transported. For details on these experiments, please see Supplementary Fig. 1e,f and the corresponding Figure legends as well as the 2nd half of the last paragraph of section “Functional characterization of SfmMCT” and the beginning of section “Ligand binding pocket description”. Considering that TSA is not transported, the terminologies “ligand” and “ K_i value” are appropriate.

- In general, this whole section is rather difficult to follow and has to be better structured. I suggest to first focus at the likely more relevant binding of lactate and then shortly discuss the TSA binding.

Authors: We have discussed and considered this suggestion, and noticed that changing the order (first discussing L-lactate and then TSA) would break the flow of the manuscript and introduce possible issues and unclarities. Therefore, and because we want to discuss the structure and the binding pocket of SfmMCT using the structure with the highest resolution, we prefer not to change the order in this section and are confident that the revised version of our manuscripts (including the improved Figs. 3 and 5) reads well. We hope for the understanding of the Reviewer.

- Additionally, it would be very helpful to add a figure, which localizes the binding site within the overall structure.

Authors: We appreciate this idea, which we have adopted and has provided more clarity regarding the location of the binding site. Please see Fig. 3a and 5a, which localizes the discussed sites in the overall structure.

- How high is the sequence identity of the binding site with the binding sites in MCT1 and MCT4. If the sequence identity with respect to the binding site is high, a supplementary figure modeling those binding sites in comparison to the given binding site would be very helpful to follow the authors’ general argumentation.

Authors: Thank you for raising this interesting point. We have provided a multiple sequence alignment of SfmMCT and human MCT1-4 in the original manuscript (Supplementary Fig. 9). Based on this sequence alignment, binding site residues share a similarity of 33% (SfmMCT vs human MCT1) and 56% (SfmMCT vs human MCT4), respectively.

Halestrap and colleagues have provided a homology model of rat MCT1 in the outward-open conformation (Wilson et al., 2009, J. Biol. Chem. 284:20011-21). On request of Reviewer 2, we compared this homology model with our SfmMCT structure, and found that the model is problematic. In particular, the positions and orientations of some key residues in the binding site differed considerably, questioning the validity of the homology model (please see above Reviewer 2 for detailed information). Therefore, we prefer to provide our SfmMCT structure to the computational/modelling experts for generating robust homology models in the near future. However, for a comparison between rat MCT1 homology model and SfmMCT, please see Supplementary Fig. 11b and the corresponding comment to Reviewer 2’s question above.

7) Proton sensor:

- In general, as for the paragraph above, I find it difficult to follow the argumentation. E.g. why are you suddenly referring to lactate binding again (line 230)?

Authors: In this sentence two positively charged amino acid residues are mentioned. To state that these two positively charged residues are not involved in the binding of the negatively charged substrate L-lactate, we made a corresponding comment in this sentence.

We have now re-phrased this sentence and hope it is better understandable, please see end of section “Proton binding site of SfmCT and pH dependence of L-lactate transport” in Results.

- To be honest, I doubt the existence of the proton sensor and I miss a proper introduction into the concept of this sensor. What would be the physiological relevance for a bacterium? Isn't the proton sensor rather important for the human homologs, which seem to fulfill different jobs probably even in the same tissue? The histidin is not even conserved... Instead, I hypothesize that the residues that show the strongest effect upon mutating (no transport) are actually involved in proton binding for its co-transport.

Authors: The suggestion proposed by the Reviewer is interesting and makes sense. Considering this feedback, we adopted the expression “proton binding site” instead of “proton sensor” in the revised version of our manuscript. The authors feel that this change has improved the clarity of this section.

- Perhaps the mutations causing increased uptake at alkaline conditions actually changed the coupling of lactate to protons transported...

Authors: We agree with this suggestion and have introduced it, please see 3rd paragraph of Discussion.

- A figure that localizes the discussed ‘sensor’ area within the overall structure would be helpful.

Authors: As for other Figures, we have adopted this suggestion, which provides clarity on the location of the proton binding site (Fig. 6a).

- If the authors disagree with my hypothesis, where do you expect protonation/deprotonation for the co-transport? I am missing a clear separation of both aspects.

Authors: As stated in the previous three points, the authors fully agree with this hypothesis and expect that protonation/deprotonation for the co-transport occurs in the proton binding site.

- What are the data in figure 6C normalized to?

Authors: The curves shown in Fig. 6d (previously Fig. 6c) represent merged data from three independent experiments. The average of the three experiments was calculated as follows:

i) Fitting a sigmoidal curve to the raw data of each individual experiment, i.e., counts-per-minute (cpm) vs pH. ii) Cpm values of each experiment were normalized with respect to the determined upper plateau value, i.e., fitted upper plateau value corresponds to 100%. iii) The arithmetic average of the three independent experiments was calculated and is represented in Fig. 6d. In the Methods section of the revised version, we have extended this description to make it better understandable.

- Why start the wildtype and K377A not at 100%

Authors: The reason for this is the described fitting procedure (see previous point). The presented transport through SfmCT and SfmCT-K377A do not reach the fitted maximum/plateau (i.e., 100%) at the lowest tested pH, i.e., pH 6. Since we wanted to use the same buffering agent (i.e., Bis-Tris propane) for all pH values to exclude any possible effects of different buffering agents, it was not possible to measure at lower values than pH 6.

8) The authors claim that they “provide a framework for understanding the molecular working mechanisms of substrate binding and transport of SLC16 transporters” (lines 254f). I disagree to this statement. They only

described the binding site for substrate while neither discussing about the coupling nor the induced conformational changes, which is required for a transport MECHANISM.

Authors: We agree with the Reviewer that our statement was too strong. Therefore, we have moderated our statement by removing “transport” in that sentence. New sentence: “Our ligand-bound S_fMCT structures and functional studies provide a framework for understanding the molecular working mechanisms of substrate binding of SLC16 transporters”.

REVIEWERS' COMMENTS:

Reviewer #1 (Remarks to the Author):

I believe the authors have significantly improved the readability of their manuscript and with all additional experiments the results are more convincing now. The only minor remark, I would add the exact % for sequence identity in 'GlpT shares low sequence identity with rat MCT1 and MCT4...' to make the statement stronger. Otherwise it has become a nice and interesting story.

Reviewer #2 (Remarks to the Author):

Thank you for revising the manuscript and answering to my satisfaction all my questions and concerns. Congratulations!
Ulrich Schweizer

Reviewer #3 (Remarks to the Author):

The authors have addressed all my questions and concerns in the revised version of the manuscript, which in my opinion improved the paper significantly. I congratulate the authors with this nice work. I have only three minor comments that should be corrected before publication:

1. lines 190ff: "The correct register of the built model was confirmed by the anomalous difference densities of selenium atoms obtained by SAD phasing of selenomethionine substituted SfMCT crystals (Supplementary Table 1 and Supplementary Fig. 3a)." Did you really use the Sel-Met crystals for phasing? Please clarify or delete "by SAD phasing" in this sentence.
2. line 200: this should be fig. 2b instead of 1b
3. Suppl. Fig 1b: wrong label of the y-axes as different radioactive compounds were used

Point-by-point discussion to Bosshart *et al.* NCOMMS-18-34613A

Reviewer #1 (Remarks to the Author):

I believe the authors have significantly improved the readability of their manuscript and with all additional experiments the results are more convincing now. The only minor remark, I would add the exact % for sequence identity in 'GlpT shares low sequence identity with rat MCT1 and MCT4...' to make the statement stronger. Otherwise it has become a nice and interesting story.

Authors: We thank the Reviewer for the positive comments. The sequence identity values between GlpT and rat MCT1 and MCT4 are provided in the revised version of the manuscript at the corresponding position: "... and GlpT shares low sequence identity with rat MCT1 and MCT4 (i.e., 18.5% and 17.3%, respectively)."

Reviewer #2 (Remarks to the Author):

Thank you for revising the manuscript and answering to my satisfaction all my questions and concerns. Congratulations!
Ulrich Schweizer

Authors: Thank you, we highly appreciate this positive feedback.

Reviewer #3 (Remarks to the Author):

The authors have addressed all my questions and concerns in the revised version of the manuscript, which in my opinion improved the paper significantly. I congratulate the authors with this nice work.

Authors: Thank you, we highly appreciate this positive remark.

I have only three minor comments that should be corrected before publication:

1. lines 190ff: "The correct register of the built model was confirmed by the anomalous difference densities of selenium atoms obtained by SAD phasing of selenomethionine substituted SfMCT crystals (Supplementary Table 1 and Supplementary Fig. 3a)." Did you really use the Sel-Met crystals for phasing? Please clarify or delete "by SAD phasing" in this sentence.

Authors: Thank you for this comment. Following the advice, we have deleted "by SAD phasing".

2. line 200: this should be fig. 2b instead of 1b

Authors: Thank you for this note. We have corrected the link to the corresponding figure in the revised version of the manuscript.

3. Suppl. Fig 1b: wrong label of the y-axes as different radioactive compounds were used.

Authors: Thank you for finding this mistake. We have updated the label of the y-axis in Supplementary Figure 1b in the revised version of the manuscript.